# AI for Global Climate Cooperation: Modeling Global Climate Negotiations, Agreements, and Long-Term Cooperation in RICE-N

**Tianyu Zhang** [* 1 2] **Andrew Williams** [* 1 2] **Phillip Wozny** [* 3] **Kai-Hendrik Cohrs** [* 4]
**Koen Ponse** [5] **Marco Jiralerspong** [1 2] **Soham Rajesh Phade** [6] **Sunil Srinivasa** [7] **Lu Li** [8]
**Yang Zhang** [9] **Prateek Gupta** [10] **Erman Acar** [11] **Irina Rish** [1 2 12] **Yoshua Bengio** [1 2 12] **Stephan Zheng** [13]

## Abstract

Global cooperation on climate change mitigation is essential to limit temperature increases while supporting long-term, equitable economic growth and sustainable development. Achieving such cooperation among diverse regions, each with different incentives, in a dynamic environment shaped by complex political and economic factors, without a central authority, is a profoundly challenging game-theoretic problem. This article introduces RICE-N, a multi-region integrated assessment model that simulates the global climate, economy, and climate negotiations and agreements. RICE-N uses multi-agent reinforcement learning (MARL) to incentivize agents to develop strategic behaviors based on the environmental dynamics and the actions of others. We present two negotiation protocols: (1) Bilateral Negotiation, an example protocol and (2) Basic Club, inspired by Climate Clubs and the carbon border adjustment mechanism (Nordhaus, 2015; Commission, 2022). When we compare their impact against a no-negotiation baseline with various mitigation strategies, we find that both protocols significantly reduce temperature growth at the cost of a minor drop in production while ensuring a more equitable distribution of the emissions reduction costs.

## 1 Introduction

The latest Intergovernmental Panel on Climate Change (IPCC) report emphasizes the urgent need for immediate action, warning that it is "now or never" to avert a climate disaster (Pörtner et al., 2022). Ecosystems are drastically changing: the Amazon rainforest is receding (Lovejoy & Nobre, 2018) and polar ice sheets are melting (Boers & Rypdal, 2021; DeConto et al., 2021). Extreme weather events, including the recent increase in coastal flooding and forest fires are unequivocal warning signs (Kundzewicz, 2016; Schmidt et al., 2022). These developments are increasingly being attributed to climate change and driving towards a system-wide tipping point (van Oldenborgh et al., 2021; Stott et al., 2016).

Climate change is a global issue impacting all. In response, public and private financing have driven technological innovation (e.g. in renewable energy) and community initiatives for systemic change. However, mitigation investments vary across countries due to social and economic factors. For example, developing nations may prioritize basic needs, while developed nations likely have more resources to address climate impacts.[1] This creates a "tragedy of the commons," where self-interest can lead to harmful outcomes for all (Gardiner, 2001).

As such, achieving and maintaining global cooperation is crucial to achieve the Paris Agreement's long-term goal of limiting the global temperature rise above pre-industrial levels to well below $2°C$ (DeConto et al., 2021). At the same time, it is important to maintain economic development, including growth and inequality reduction. For instance, international trade treaties, foreign investment, and technology transfer can help developing countries meet net-zero targets while supporting global economic growth. Such cooperation could be fostered through *climate clubs*, which tackle barriers to climate action (Nordhaus, 2015).

From a modeling perspective, achieving and maintaining global cooperation poses a complex game-theoretic problem involving cooperation, communication, and competition. It can be modeled with $n$ strategic agents, each representing a region or nation seeking to maximize their own utility through policies aimed at their own socio-economic and

---

[*]Equal contribution [1]Mila - Quebec AI Institute [2]DIRO, Université de Montréal [3]Vrije Universiteit Amsterdam [4]Universitat de València [5]Leiden University [6]Wayve [7]NVIDIA [8]University of Pennsylvania [9]Bank of Canada [10]University of Oxford [11]University of Amsterdam [12]CIFAR [13]Asari AI. Correspondence to: Tianyu Zhang <tianyu.zhang@mila.quebec>.

*Proceedings of the 42nd International Conference on Machine Learning*, Vancouver, Canada. PMLR 267, 2025. Copyright 2025 by the author(s).

---

[1]Developing countries typically pay more to reduce emissions than higher-income countries (Erickson et al., 2015).

climate goals, which may conflict with those of the other agents. These agents interact through trade, diplomacy, or foreign aid and investments, with cooperation manifesting through mutual negotiation and agreements.

A key issue is the lack of central authority to enforce cooperation or compliance with the agreements in the real world. Therefore, it is essential to design negotiations and agreements that promote *sustained* cooperation in mitigating climate change while allowing all parties to achieve their individual policy goals.

Such game-theoretic problems present unresolved technical challenges. For instance, a key analysis in the 2022 IPCC report predicts climate change under five different so-called Shared Socio-Economic Pathways (SSPs), each based on a set of predefined climate-economic policies for each global region (Pörtner et al., 2022). However, a key limitation is that it is unclear whether these policies would be implemented by utility-maximizing actors, making it uncertain how likely these scenarios are to occur or how robust they are to changes in agent behavior over time.

Climate-economic policy tradeoffs are typically modeled using integrated assessment models (IAMs), which quantify the effects of economic activity and $CO_2$ emissions on global temperatures and long-term economic development. A pioneering example is the Dynamic Integrated model of Climate and Economy (DICE) from (Nordhaus, 2007), which models the links among climate and economic factors, such as population growth, technological change, $CO_2$ emissions, global temperatures, and economic damages. DICE uses a single global economy. The Regional Integrated Model of Climate and Economy (RICE) extends DICE to multiple regions (Nordhaus & Yang, 1996b) and can include tariffs and trade (Nordhaus, 2015; Lessmann et al., 2009). While widely used, RICE has its limitations, such as unrealistic assumptions, lack of distributional analysis, simplistic definition of regional interaction, and no consideration of uncertainty (Pindyck, 2013; Farmer et al., 2015; Gazzotti, 2022). Thus, RICE needs significant modification in order to capture the strategic behavior in climate negotiations. To address these issues, we draw inspiration from agent-based modeling (Bonabeau, 2002) as a bottom-up modeling framework and advances in MARL to identify effective policies (Zheng et al., 2022a) and train strategic agents (Silver et al., 2016; Vinyals et al., 2019).

Our main contributions are as follows:

**RICE-N Integrated Assessment Model** We introduce RICE-N, an integrated climate-economic model based on RICE (Nordhaus & Yang, 1996a). Designed as a simulation tool for climate negotiations, RICE-N incorporates MARL to model realistic interactions between agents. Furthermore, RICE-N is modular, allowing flexibility to accommodate various climate-economic model configurations.

**Negotiation Protocols** As RICE-N is built for modeling negotiations, we develop two novel protocols: (1) Bilateral Negotiation, a baseline protocol that provides simple inter-agent communication; (2) Basic Club, a protocol inspired by actual climate economic policy (Nordhaus, 2015; Commission, 2022) to foster burden sharing among agents.

**Analyzing Results** To evaluate the Basic Club and Bilateral Negotiation protocols, we compare the climate-economic performance of agents trained with and without these negotiation protocols. Metrics include inequality in emission reduction costs as %GDP and emissions. Compared to agents trained without negotiation protocols, agents trained under the negotiation protocols reduce temperature growth at the cost of a minor drop in production while ensuring a more equitable distribution of the emissions. Our code samples are available here: https://github.com/mila-iqia/climate-cooperation-competition

## 2 Related Work

Previous work has studied climate change through various lenses: political economy and negotiations (Chan et al., 2022; Bakaki, 2022), public perception and institutional dynamics (Moore et al., 2022), and coalition formation (Zenker, 2019). Although IAMs have been used to study the impacts of political negotiation (Rochedo et al., 2018), the game-theoretic aspects of climate cooperation using machine learning and calibrated IAMs remain unexplored.

Climate negotiations have been analyzed using game theory, from simple prisoner's dilemma models (DeCanio & Fremstad, 2013) to more complex bargaining games with learning (Smead et al., 2014; Greeven et al., 2016). However, these simplified models often lack crucial real-world features like multilateral settings, strategic behavior with multiple goals, evolving dynamics, and agent heterogeneity (Madani, 2013).

Recent advances in multi-agent reinforcement learning (MARL) offer new approaches to studying strategic behavior in complex environments (Shoham & Leyton-Brown, 2008). While MARL has been applied to climate-related problems like HVAC optimization (Mai et al., 2024) and theoretical cooperation studies (Jaques et al., 2019), it has not yet been extended to rich, calibrated climate-economic simulations. Our work addresses this gap by combining MARL with detailed climate-economic modeling. For an extensive review of related work, please refer to Appendix A.

## 3 The RICE-N Integrated Assessment Model

We introduce RICE-N, an IAM that further augments RICE with a framework for negotiation protocols, and also includes international trade and tariffs, following (Lessmann et al.,

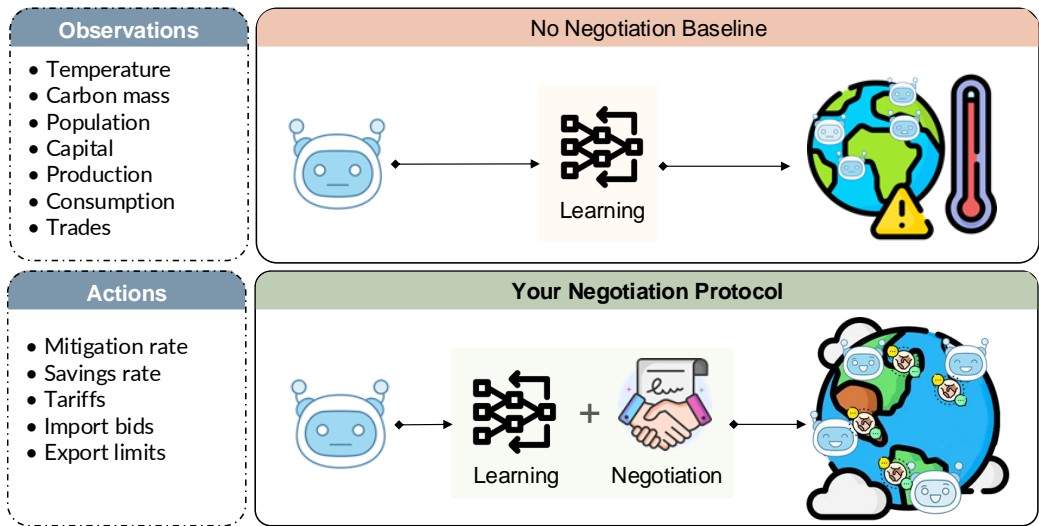

*Figure 1.* **Schematic overview of how negotiation can lead to better outcomes in RICE-N.** Each region (agent) uses a policy model to make climate, economic, and trade decisions. For clarity, we show the flow of information and actions for a single agent only. Agents can negotiate, but a negotiation protocol must be implemented, such as the example negotiation protocols in Section 4. At a high-level, at each timestep, each policy chooses climate and economic actions based on the observations they receive. (TOP) If there is no negotiation protocol, agents immediately decide their actions, which leads to high climate damage as can be seen in Figure 3. (BOTTOM) If a negotiation protocol has been implemented, agents must first negotiate before performing actions. The outcome of all negotiations is agreements (or lack thereof), which may be between two or more agents. In particular, an agreement may influence the remaining actions that an agent can take in the climate and economic domains. For the same timestep, each agent then makes decisions with respect to the climate, economy, and trade.

2009)). As such, RICE-N shares climate, economic and social characteristics with the real world.

In RICE-N, there are $n$ regions, each modeled as an independent decision-making agent. Regions interact with each other and the environment through their actions: setting a savings rate, mitigation rate, trades and tariffs, and negotiation actions, for each time step.

RICE-N has two main components: *negotiation* and *climate-economic activity*, see Figure 1. The activity component simulates the physical actions of the agents and the resulting evolution of the environment. The negotiation component simulates communication between regions, allowing them to influence each other's behaviors and form agreements. Agreements may, in turn, adjust the available actions for each region during the activity stage.

Each simulation episode consists of $H$ steps, each representing $\Delta$ years (e.g, $\Delta = 5$). Thus, the simulation lasts for $H \times \Delta$ simulation-years. At every step, the simulation goes through the negotiation stages, and agreements are formed between regions. The simulation then enters the activity stage where each region takes actions that are affected by the agreements formed during the negotiation stages.

**Climate-economic dynamics overview.** The *state of the world* is characterized by global variables such as the

concentration of $CO_2$ levels in the Earth's atmosphere, and the average global temperature, as well as region-specific variables such as population, capital, technology level, carbon intensity of economic activity, and balance of trade. For more details, see Table 2 in Appendix B for variables and Appendix H for calibration details.

RICE-N has climate and economic dynamics. The *climate dynamics* model how $CO_2$ levels in the atmosphere impacts global temperatures. The *economic dynamics* model how technology levels, capital, population, and gross domestic production evolve. Notably, the climate dynamics impact the economic dynamics through a *damage function*, which describes how higher temperatures lead to losses in capital.

These dynamics depend on savings and mitigation rates set by each agent, e.g., agents may choose to invest more in climate change mitigation, but this may lower economic productivity in the short-term.[2] As global $CO_2$ levels and temperatures affect all agents, these dynamics mean

---

[2]In economic terms, variables such as capital, balance of trade, carbon mass, and global temperature depend on the agents' actions and are called *endogenous* variables. On the other hand, variables such as population, technology level, and carbon intensity of economic activity are called *exogenous*, i.e., their values do not depend on the agent's actions. Note that the values of exogenous variables can vary across steps in a predetermined manner.

the decisions of each agent affect the climate-economic outcomes for other agents, too.

The activity component encapsulates these dynamics. For each step: 1. The gross output production for each region is computed based on the state of the region, in particular, its capital investment, labor (or population), and technology factor. 2. The net economic output is the gross output production reduced by climate damages from rising global temperature, and the cost of efforts towards mitigation by this region. 3. The region consumes domestic goods equal to the quantity of the net economic output that is left after capital investment and export. It also consumes foreign goods from imports. 4. The consumption utility for each region from consuming domestic and foreign goods is computed using the Armington elasticity assumption that has become standard in international computable general equilibrium models (Armington, 1969). This gives the *reward* corresponding to each region in every step. For more details, please refer to Appendix C.1 and Table 3 in Appendix B.

**International trade and tariffs.** RICE-N features international trade to exchange and transfer goods between agents, following (Lessmann et al., 2009). Here, agents are modeled to seek diversity in their consumption according to the Armington assumption (Armington, 1969), so they want to consume goods produced by other agents and are willing to export some of their own goods in exchange. Each agent specifies the amount of goods it wishes to import and sees its orders filled partially or fully, depending on the other agents' willingness to export. In addition, agents can choose to impose import tariffs to restrict trading. Import tariffs restrict the consumption of goods to which they are applied, implicitly increasing their prices. These price increases make imported goods without import tariffs more attractive. Therefore, trade and tariffs force agents to engage with other regions and be strategic, incentivizing negotiations and agreements. See Appendix C.2 for more details.

**Extensions** The climate dynamics of DICE and RICE are based on (Nordhaus, 2018; Kellett et al., 2019). Economic analysis based on these dynamics suggests that optimal policy paths limit global warming to $3.5°C$ and lead to net zero economies in the next century (Nordhaus, 2019). Several works have criticized the trustworthiness of these results due to shortcomings in the model, such as missing representations of climate risk and uncertainty (Daniel et al., 2019), unrealistic damage functions (Drupp & Hänsel, 2021) and oversimplified climate dynamics (Mattauch et al., 2020). Recent work has shown that updating the model with a more realistic climate emulator and recent damage estimates results in optimal trajectories that are more aligned with the climate targets of the Paris agreement (Hänsel et al., 2020).

RICE-N's modularity enables us to easily update the equations governing the dynamics, enabling comparison to existing studies with various setups. In particular, the climate, economic and trade components are loosely coupled to enable isolated extensions and modifications. The order in which the dynamics act on the world state is determined in the global `climate_and_economy_step()` function, which can accommodate extensions or modifications with minimal changes. For example, similar to (Hänsel et al., 2020), we provide an updated and higher damage function based on a recent meta analysis (Howard & Sterner, 2017). We also implement the climate dynamics of the famous *Finite Amplitude Impulse Response (FaIR)* model, an emissions-based climate model that is also featured in the IPCC reports (Millar et al., 2017).

In Figure 2, we analyze different damage and climate functions. We compare three baselines: agents that always perform the *minimum* mitigation, the *maximum* mitigation and agents trained with *no negotiation*. We see that the effect of the damage function is marginal on the strategy of the agents, and even higher damage does not steer them away from maximizing their own utility. The choice of climate dynamics clearly has an effect on the resulting possible trajectories. The lowest emissions scenario for the DICE dynamics limited warming to between $2.6°C$ and $2.7°C$ by the end of the century. In contrast, the updated FaIR dynamics show that $2°C$ is still possible, which corresponds roughly to the low GHG emission scenario (SSP1-2.6) of the IPCC (Pörtner et al., 2022). As DICE2016 remains the most ubiquitous model in use, we continue with the DICE2016 climate and damage functions for the remainder of the work.

## 4 Negotiation Protocols

A negotiation protocol is a communication channel through which agents can make promises and demands that ultimately constrain their behavior. We design RICE-N to be modular such that its base dynamics can be used to test a variety of different negotiation protocols.

In principle, a negotiation protocol can have many desirable properties, including but not limited to the following (Nisan et al., 2007; Luo et al., 2024):

- *Incentive-compatibility*: Agents have an incentive to act according to their true preferences. (Pavan et al., 2014),

- *Self-enforcing*: No external body or agent is required to enforce other agents to participate in negotiations and adhere to agreements (Telser, 1980),

- *Fairness*: Agents get comparable utilities (Luo et al., 2024).

Negotiation steps occur prior to the climate and economic step so that agents' climate-economic behavior is influenced by the outcome of negotiations through binding commitments.

RICE-N is flexible enough to serve as a basis for different

negotiation protocols. We implement two negotiation protocols: *Bilateral Negotiaton* and *Basic Club*. The former serves as an example to illustrate the key phases of negotiation, and the latter is modeled on actual climate economic policy.

**Bilateral Negotiation**     In addition to the base dynamics of RICE-N, Bilateral Negotiation agents perform the following steps:

- *Proposal stage*: At this stage, each agent $i$ makes a proposal, $(\hat{\mu}_i, \hat{\mu}_j)$, to every other agent $j$ where $\hat{\mu}_i$ indicates the mitigation level that agent $i$ promises and $\hat{\mu}_j$ what agent $i$ requests from agent $j$.

- *Evaluation stage*: At this stage, each agent observes the proposals made to it in the preceding stage and takes an action of accepting or rejecting each of the proposals.

- *Commitment stage*: At this stage, each agent commits to the maximum mitigation rate of all accepted proposals.

**Basic Club**    Basic Club is a multi-agent negotiation protocol based on climate clubs and the Carbon Border Adjustment Mechanism (CBAM) (Nordhaus, 2015; Commission, 2022). The former was initially proposed by William Nordhaus and codified by Article 6 of the Paris Agreement; a climate club is a coalition of regions that agree to a common

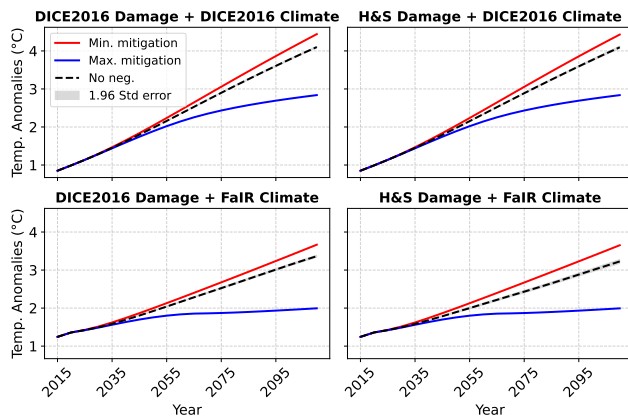

*Figure 2.* Extensions to the base setup of the framework. RICE-N is highly adaptable in that different components, such as the climate model or damage function, can easily be exchanged. We compare the basic no-negotiation baseline (see Section 6) over different damage functions and climate dynamics. The *DICE2016 damage function* and *DICE2016 climate model* are taken from (Nordhaus, 2018). The *H&S damage function* is based on the recent estimates of (Hänsel et al., 2020), and the *FaIR climate model* is a famous emission-based climate emulator (Millar et al., 2017). The H&S damage function punishes temperature increases more strongly than the DICE2016 damage function, but the effect is still too mild during the roll-out period to lead to significant behavior changes in the agents under the no-negotiation scenario. The FaIR climate model implements more realistic climate dynamics and shows that a path to $2°\text{C}$ is still possible.

emission reduction target and impose a uniform tariff on all goods coming from non-club members (Nordhaus, 2015; 2021b). The latter (i.e., CBAM) is a feature of the EU Green Deal, aiming to address carbon leakage and facilitate net zero emissions by 2050 (Commission, 2020; 2022). Basic Club borrows the ideas of a uniform tariff on all goods and a variable tariff value that depends on the emission reduction target of the exporting country from Nordhaus and CBAM, respectively (Nordhaus, 2015; Commission, 2020; 2022). The legality of Basic Club with respect to World Trade Organization compliance is described in Appendix J.

Formally, the Basic Club functions as follows:

- *Proposal*: Each agent $i \in \{1, ..., n\}$ proposes $\hat{\mu}_i \in [0, 1]$, indicative of the mitigation rate of the club they would like to join.

- *Evaluation*: Each agent $i \in \{1, ..., n\}$ evaluates proposals $\hat{\mu}_j$ from other agents $j \neq i$, by either accepting or rejecting them. Let $A_i$ be the set of mitigation rates of accepted proposals for region $i$.

- *Club Formation*: We define the minimum mitigation rate of agent $i$ as $\mu_i = \max A_i$ if $A_i$ is not empty, otherwise, it is 0. A club $c$ is a subset of agents with a common minimum mitigation rate $\mu_c$.

- *Sanctions*: For each club $c$, non-members with lesser minimum mitigation rates the club mitigation rate i.e., $\mu_j < \mu_c$ receive a tariff as the differences in rates i.e., $\tau_{j,c} = \mu_c - \mu_j$.
  Members and non-members with $\mu_j \geq \mu_c$ have $\tau_{j,c} = 0$.

Exact descriptions of the action spaces for both protocols can be found in Appendix C.3.

**Binding Commitments**     Commitments are made binding through action masks, which control the accessible action space during the following step. For example, negotiation could yield a binding commitment to a minimum of 20% mitigation rate. Then, the action mask only allows setting mitigation rates above that level for this region. We discuss extensions beyond this paradigm in Section 7.

In Section 6, we will compare the climate-economic impacts of Basic Club, Bilateral Negotiation and No Negotiation.

## 5 Modeling Strategic Agents using MARL

The negotiation protocols and the climate-economic dynamics of RICE-N define a game-theoretic setup between the different regions. We model the behavior of an agent $i$ using its policy $\pi_i(a_t | o_t)$ that maps the agent's observations $o_t$ to a probability distribution over its actions $a_t$ at time $t$.

**Reason to use MARL**     Existing IAMs often use a predetermined policy for each agent that is fixed exogenously. Such approaches would require handcrafted agent policies

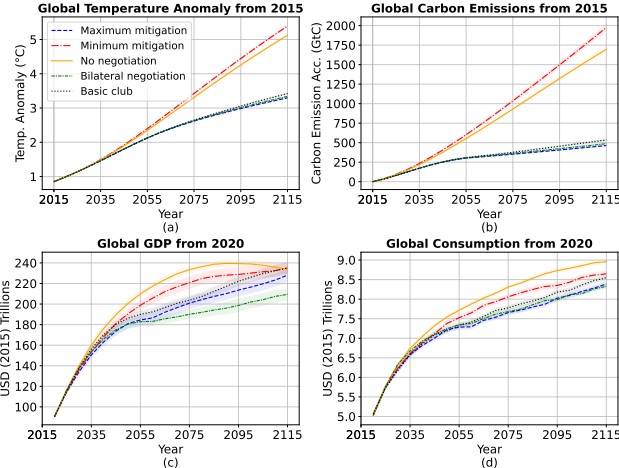

*Figure 3.* Comparison of climate-economic outcomes among different baselines (*minimum mitigation, maximum mitigation*, and *no negotiation*) against negotiation protocols (*Bilateral negotiation* and *Basic Club*) with mean ±1.96 stderr (see Section 6 for more details). In respective order, we show the comparison of (a) global temperature anomaly (lower is better), (b) global carbon emissions (lower is better), (c) global output production (higher is better) and (d) global consumption (higher is better). While *no negotiation* performs best in the early stages of the simulation, it exhibits a downward trend after 2090, suggesting that the adverse effects of rising temperatures start to outweigh the economic benefits. In contrast, *Basic Club* surpasses *no negotiation* economically towards the end of the simulation while successfully maintaining a lower increase in temperature anomaly and carbon emissions.

for each different negotiation protocol. Furthermore, the reliability of the outcomes from the simulation would depend on the modeled policies.

In contrast, we assume that each region strategically interacts with the environment and other strategic agents in it. Therefore, instead of manually setting the behavioral policy $\pi_i$, we use machine learning techniques to find policies that seek to maximize the objectives of the agents, and hence derive the agent policies endogenously.

Specifically, the agents are assumed to be utility-maximizing such that each agent $i$ optimizes its policy $\pi_i(a_t|o_t)$ to maximize its long-term aggregate $\gamma$-discounted utility:

$$\max_{\pi_i} \mathbb{E}_{\pi_1,\dots,\pi_n} \left[ \sum_{t=0}^{H} \gamma^t r_{i,t} \right], \quad (1)$$

where $r_{i,t}$ is the utility of the region $i$ at step $t$ determined by its *aggregate consumption* $C_{i,t}$ as follows:

$$r_{i,t} = U_{i,t} = \frac{w}{1-\alpha} L_{i,t} \left( \left( \frac{C_{i,t}}{L_{i,t}} \right)^{1-\alpha} - 1 \right), \quad (2)$$

where $L_{i,t}$ is the *population* of the region $i$ at step $t$; $w$ is the *welfare loss multiplier* that decreases the utility that a given

region receives proportionally to how other regions tariff that region's exports (Nordhaus, 2021a) (see Appendix G for more details); The parameter $\alpha \geq 0$ is the *consumption elasticity* which can represent the degree of risk aversion or the (un-)willingness of society to sacrifice consumption today for consumption in the future. The *discount factor* $\gamma$ models the long-term value of rewards for the agents, and as such it could differ across different agents. The discount factor for each region is often updated with the changing administration; in the absence of any consensus, we fix $\gamma$ to be homogeneous across agents as assumed in (Nordhaus, 2015). The aggregate consumption $C_{i,t}$ in the above equation is obtained by combining the domestic and foreign goods consumption using the Armington model (Armington, 1969):

$$C_{i,t} = \left( \psi^{\mathrm{dom}}(C_{i,i,t})^\lambda + \sum_{j \neq i} \psi^{\mathrm{for}}(C_{i,j,t})^\lambda \right)^{\frac{1}{\lambda}}, \quad (3)$$

$$C_{i,j,t} = x_{i,j,t}(1 - \tau_{i,j,t}) \quad \forall j \neq i, \quad (4)$$

where $C_{i,j,t}$ is the foreign goods consumed after imposing tariffs $\tau_{i,j,t}$ on the imported goods $x_{i,j,t}$ by region $i$ from region $j$ at step $t$. $\psi^{\mathrm{dom}}$ and $\psi^{\mathrm{for}}$ are shared parameters, and $\lambda$ is the Armington elasticity parameter that represents the degree to which consumers are willing to switch between domestic and imported goods when prices change.

**Multi-agent reinforcement learning (MARL)**  Simulating a utility-maximizing agent requires computing the optimal policy for each agent[3]. Finding the optimal utility-maximizing policy for each agent in response to complex environment dynamics and other agent policies naturally leads to MARL; (Busoniu et al., 2008) provides a comprehensive overview of the topic. In short, MARL extends single-agent RL to find an optimal policy for each agent interacting in a dynamic environment to solve Equation 1. The RL framework models how an agent's actions affect the state of the environment and its rewards (utilities). Thus, an RL agent has to learn to anticipate the long-term effects of its actions. This is especially true and challenging in multi-agent environments such as RICE-N, where agent actions affect key climate-economic metrics including global temperatures, capital investments, carbon emissions, etc. In addition, MARL algorithms have to deal with the additional game-theoretic challenge of each agent's response to the policies of other agents. This makes the task of finding the optimal policy a moving target (until a form of equilibrium is reached in the agent policies).

**Equilibrium Concept**  In contrast to the original RICE IAM (Nordhaus & Yang, 1996b), which discusses pure strategy Nash equilibria, RICE-N employs reinforcement

---

[3]Although we assume the agents to be utility-maximizing, other behavioral models could also be implemented in RICE-N.

learning agents operating in a dynamic, partially observed environment with expanded action spaces and evolving negotiation protocols. As such, there is no single, fixed equilibrium concept that applies across all scenarios modeled in RICE-N. Instead, the emergent behavior of agents can approximate different forms of equilibria depending on the structure of the negotiation protocol:

1. Equilibrium type: The negotiation protocol can give rise to the introduction of previously irrelevant equilibrium concepts, such as correlated equilibria from a stochastic protocol.

2. Punishment and enforcement: Action spaces that include tariffs or sanctions can affect the equilibrium and introduce the possibility of self-enforcement through collective punishment for defection.

3. Non-binding agreements: If commitment masks are relaxed to allow cheap talk, other solution concepts such as coalition-proof Nash equilibria become relevant (Bernheim et al., 1987).

4. Information structure: The negotiation protocol can affect what information is public vs private.

Thus, equilibrium behavior in RICE-N is not defined a priori, but emerges from the interaction of learning dynamics, available actions and the design of the negotiation process.

**Implementing RL Agents**    Our code includes both CPU and GPU implementations of the full RL pipeline using A2C (Mnih et al., 2016). RICE-N can also be used with other RL implementations. Our base implementation models each RL agent using a neural network policy that shares weights across agents, but uses agent-specific inputs. The architecture of the network can be adjusted, e.g., the number of layers and the dimension of each layer. Agent policies use separate heads for each action. To distinguish between agents, the policy model's input contains agent-specific features, e.g., their population, capital, technology factor, damage function, and a one-hot representation of the region's index, as well as the public state of the world (e.g., climate conditions). In addition, each agent receives information about negotiations, e.g., the latest proposals made to and by this agent, or the minimum mitigation rate agreed upon by this agent. Depending on the negotiation protocol, not all observations and actions are relevant to each agent. How negotiations evolve depends on the specifics of the protocol and the different actions executed by the agent, e.g., proposals for other agents, decisions on proposals made by other agents, and setting mitigation and savings rates that may or may not be in line with what was agreed upon.

## 6    Evaluating Negotiation Protocols

In this section, we use RICE-N to analyze the climate and economic outcomes of the *basic club* and *bilateral negotiation* protocols (see Section 4). We compare the climate-economic outcomes of these protocols with three baselines, namely a minimal mitigation policy, a maximal mitigation policy, and a mitigation with no negotiation policy. We also examine the (group) fairness of these protocols across regional contributions by calculating the Gini Index of various climate-economic variables measured by RICE-N.

**Experimental Setup**    To compare outcomes with and without negotiation, we train five models consisting of (i) Basic Club, (ii) Bilateral Negotiation, (iii) a no negotiation baseline, (iv) Maximum mitigation, and (v) Minimum mitigation. The latter two models can only mitigate either the maximum or minimum possible amount, respectively; all other actions in (iv) and (v) are trained as normal. Each model is trained for $30,000$ episodes. For evaluation, we gather 50 rollouts of each model using a unique seed per run to ensure a varied distribution of outputs. For more details about the concrete time series, please refer to Appendix M.

**Negotiation Protocols Can Improve Climate-Economic Outcomes.**    In Figure 3, we illustrate the global temperature anomaly, carbon emissions, output production and consumption across time steps. To establish upper and lower bounds in performance, we also compare our findings to maximal and minimal mitigation strategies. The former consists of the maximum possible emissions reductions per time step and the latter features no emissions reduction.

We see that Basic Club and Bilateral Negotiation perform vastly better than the minimal mitigation and no negotiation baselines, with temperature increase and carbon emission outcomes nearly matching the maximal mitigation baseline.

Economically, however, Bilateral Negotiation falls relatively short while Basic Club is not significantly worse than the minimum mitigation, maximum mitigation and no negotiation baselines at the last time step. In fact, Basic Club ends with steady output growth, as opposed to the slowing economic growth of the minimum mitigation and no negotiation baselines. The distribution of mitigation strategies under different negotiation protocols is visible in Appendix L.

For global consumption, we see that no negotiation performs best, but this trend is unlikely to continue considering the decrease in economic production observed in later time steps because the damage from the increasing temperature would significantly limit the sustained economic growth in the long term. Minimum mitigation performs second best, but Basic Club results in nearly as much consumption at the last time step, which, paired with its promising steady growth in economic output, looks to result in more sustainable outcomes in the long run.

Overall, the large room of improvement in the global temperature levels at the expense of relatively much smaller difference in production output shows the value of these

protocols. To assess the robustness of our results, we conduct a sensitivity analysis on selected parameters based on economic theory, which are the discount factor, welfare loss weight, consumption substitution rate, and relative preference for domestic goods. The results, discussed in Appendix K, confirm that our findings over differnt scenarios are stable under changes in critical model parameters.

**Negotiation Protocols Impact Fairness**    To measure the (group) fairness of outcomes across climate economic variables of interest, we use the Gini index (Gini, 1912), a statistical measure used to quantify the inequality of variables $x$ within a population. The Gini Index ranges from 0 (perfect equality) to 1 (absolute inequality) on the variable of interest

$$G(x) = \frac{\sum_{i=1}^{n} \sum_{j=1}^{n} |x_i - x_j|}{2n^2 \bar{x}} \qquad (5)$$

where $n$ is the number of agents, and $\bar{x}$ represents the mean.

Table 1 (left) illustrates the degree of inequality of different negotiation protocols across regions with respect to abatement cost, mitigation rate, carbon emission and consumption. We first note that both Basic Club and Bilateral negotiation result in lower inequality than No Negotiation when it comes to abatement cost, mitigation rate and carbon emissions. That is, the economic burden of emission reduction is more equitably shared when agents have the opportunity to negotiate with one another about their emissions reduction targets. The inequality of carbon emissions itself is less impacted, as the carbon intensity is a largely region specific parameter. However, for the mitigation rate, the Bilateral Negotiation seems to have a significantly lower Gini Index than the Basic Club. Although this may appear desirable, it may not be optimal for all regions to contribute equally to mitigation efforts. In fact, in Table 1 (right), if we contrast the Global Output Production and the Global Carbon Emissions of Basic Club and Bilateral Negotiation, it seems that Basic Club achieves significantly better economic outcomes at the cost of a slight increase in temperature while maintaining a similar Gini Index for consumption.

**Discussion and Analysis**    From a climate oriented perspective, Bilateral Negotiation outperforms Basic Club. This stems from the commitment mechanism used. Bilateral Negotiation agents take the maximum of all accepted proposals and requests. Early on in training, those proposals exhibit a high degree of randomness. Hence, the maximum of random proposals is often near the maximum possible mitigation rate. We emphasize that Bilateral Negotiation is intended as an example for RICE-N users which can illustrate the basic steps of negotiation and is not a realistic negotiation protocol. While Basic Club performs comparable, it involves a much smaller proposal action space, and is designed to reflect more realistic climate policy with real-world climate negotiations (Commission, 2022; Nordhaus, 2015; Commission, 2020).

While Bilateral Negotiation has limited real-world applicability, the $O(n^2)$ communication complexity for $n$ agents is inefficient for large-scale international negotiations. Moreover, bilateral negotiations can lead to contradictory agreements with no feasible solution. This challenge underscores why most international climate negotiations opt for multilateral forums where all parties collectively discuss and agree upon terms, as exemplified by the United Nations Framework Convention on Climate Change (UNFCCC) process (Mantlana & Jegede, 2022).

From a climate justice standpoint, both the Basic Club and Bilateral Negotiation protocols evaluated in this study do not adequately consider regional differences. The high mitigation rates achieved by these protocols may not be equitable or desirable for regions whose historical emissions are typically lower than those of others.

**Potential Policy Implications**    Our findings suggest that the Basic Club protocol holds promise as a basis for real-world climate policy. However, the impacts of climate clubs and of border adjustment mechanisms depend significantly on how they are implemented. Without complementary measures such as redistribution and technology transfer, these mechanisms risk functioning as de facto carbon taxes on developing countries that are heavily reliant on carbon-intensive development pathways (Goldthau & Tagliapietra, 2022; Perdana & Vielle, 2022). That said, we caution that the outcomes observed in our framework should not be interpreted as direct predictions for real-world negotiations. Rather, RICE-N offers a simulation platform to explore, test, and compare the dynamics and consequences of alternative climate policy designs under controlled and transparent assumptions.

## 7    Limitations

RICE-N can be improved in a number of aspects. Firstly, we do not make use of regional damages and temperatures, which are captured by other RICE models. Representing regional disparities is critical for analyzing model outcomes from a climate justice perspective (Gazzotti et al., 2021). Secondly, there are longer time horizon versions of RICE currently available (Biswas et al., 2024) that extend the simulation up to 300 years. Furthermore, RICE-N does not represent damage disparities within regions (Dennig et al., 2015) which may obscure the mediating role of socioeconomic class on climate damages. Finally, the reasoning and decision-making logic behind the negotiation process remains a black box and lacks interpretability, as

Table 1. Climate-economic outcomes and Gini index at the last time step (year 2115). In **bold** are the best results. If two results are not significantly different, we bold both.

| | GINI INDEX | | | | CLIMATE-ECONOMIC OUTCOMES | | | |
| --- | --- | --- | --- | --- | --- | --- | --- | --- |
| NEGOTIATION PROTOCOL | ABATEMENT COST (%GDP) | MITIGATION RATE (%) | CARBON (GtC) | CONSUMPTION (USD trillion) | TEMP. (°C) | OUTPUT (USD trillion) | CARBON (GtC) | CONSUMPTION (USD trillion) |
| None | 0.695±0.018 | 0.405±0.018 | 0.654±0.008 | **0.529**±0.002 | 5.121±0.032 | **233.6**±3.9 | 1697.2±25.6 | **8.96**±0.04 |
| Bilateral | **0.333**±0.005 | **0.011**±0.007 | **0.565**±0.020 | 0.540±0.004 | **3.34**±0.03 | 209.4±5.5 | **487.0**±15.2 | 8.33±0.05 |
| Basic Club | **0.339**±0.006 | 0.030±0.006 | **0.579**±0.017 | 0.541±0.003 | 3.422±0.018 | **235.1**±5.0 | 534.7±9.50 | 8.55±0.05 |

it directly stems from the actions of agents trained through multi-agent reinforcement learning (MARL) algorithms.

In addition to addressing the aforementioned shortcomings, we aim to extend RICE-N in the following directions. We plan to integrate multi-level reinforcement learning (Zheng et al., 2022b) to model the negotiation protocol of the Conference of the Parties. Moreover, we aim to explore the inclusion of a welfare redistribution mechanism (Orlov et al., 2024) as a component of negotiation. Additionally, we are exploring the use of JAX (Bradbury et al., 2018) to significantly enhance the performance of RICE-N. This would enable wide-scale sensitivity analysis across simulation parameters. Furthermore, we will leverage large language models to enhance the interpretability of the negotiation process. In our current setup, agents cannot deviate from agreed-upon actions for the specified time step (5 years). While this assumption simplifies analysis and isolates the impact of the negotiation mechanism, it does not reflect the uncertainty and strategic mistrust present in real-world climate negotiations (e.g., countries withdrawing from agreements). Future work should explore the incorporation of non-binding commitments, which would allow agents to deviate from agreements and engage in strategic communication or cheap talk (Crawford & Sobel, 1982; Caparros, 2016).

## 8 Conclusion

In this paper, we introduced RICE-N, a novel integrated assessment model that combines climate-economic dynamics with multi-agent reinforcement learning to simulate global climate negotiations and agreements. RICE-N offers a flexible framework for testing various negotiation protocols and their impact on long-term climate and economic outcomes. We demonstrated the utility of RICE-N by implementing and comparing two negotiation protocols: Bilateral Negotiation and Basic Club. Our results show that both protocols can lead to improved climate outcomes compared to scenarios without negotiation while maintaining comparable economic performance. Notably, the Basic Club protocol, inspired by real-world climate policy proposals, achieved a balance between emissions reduction and economic growth that surpassed the no-negotiation baseline in the long term. Beyond this specific application, RICE-N offers value to a

range of research and policy communities. Machine learning researchers can leverage the modular RL component to benchmark different reinforcement learning algorithms in a dynamic, real-world-calibrated environment. Climate scientists can use the framework to compare alternative climate modules and damage functions under consistent economic and strategic assumptions. Governments may apply it to test the robustness of proposed climate-economic policies to strategic behavior and to anticipate potential impacts on international trade. International organizations, such as the OECD or WTO, could use the tool to analyze the distributional and economic trade-offs associated with alternative climate policy designs and negotiation strategies. It contributes to the development of more robust and equitable climate policies, supporting efforts to mitigate climate change while maintaining sustainable economic development.

## Impact Statement

The goal of this work is to produce a climate-economic model that helps foster robust, durable negotiation protocols. Tools for collective cooperation such as RICE-N can help us move toward more sustainable, fair, and long-lasting climate-economic outcomes. However, such tools can lead to unintended consequences, including the carbon footprint of using RICE-N, economic inequality due to its limitations on the applicability to the real world.

**Carbon footprint**  It is important to acknowledge that using RICE-N inevitably results in carbon emissions. Therefore, we encourage users to consider their energy usage when running experiments and offset their carbon emissions.

**Economic Inequality**  As previously discussed, economic inequality is inevitably intertwined with climate change. The consideration of approaches that address climate change should always include the economic inequalities that they could impact.

**Real World Potential and Limits**  It is important to note that predictions in RICE-N will eventually differ from actual outcomes due to inherent real-world complexity and the limitations of simulation dynamics. Therefore, decision makers should consider these limitations, and the possible gaps between simulated and real outcomes before making any policy decisions.

## Acknowledgments

T Zhang acknowledges the support from Microsoft and Samsung. P Wozny acknowledges the support of the Fiscal Institute of Tilburg. K H Cohrs acknowledges the support from the European Research Council (ERC) under the ERC Synergy Grant USMILE (grant agreement 855187). We acknowledge the support from the Canada CIFAR AI Chair Program and the Canada Excellence Research Chairs Program. We also acknowledge everyone who contributed to or joined AI4GCC (AI for Global Climate Cooperation) competition.

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

# A   Extend Related Work

**Multi-agent aspects of climate change.**     Previous work has studied the connection between political economy, negotiations, and climate change. Empirical work has found that previous climate summits have had inconsistent or too little impact (Chan et al., 2022; Bakaki, 2022). The impact of social dynamics has been studied in a stylized climate-social model, finding that public perception and institutional responsiveness are important to explain variations in emissions (Moore et al., 2022). The formation of coalitions and agreements under climate negotiations has also been studied from a game-theory perspective (Zenker, 2019). IAMs have also been used to study the impact of political bargaining on the economic burden required to meet climate targets (Rochedo et al., 2018). However, to the best of our knowledge, no work has analyzed the game-theoretic aspects of climate cooperation using machine learning and calibrated IAMs.

**Social dilemmas.**     Our work also is also one of the social dilemmas, situations where selfish agents act to lead to collective outcomes. These dilemmas are also a subset of general-sum games. Prominent examples of social dilemmas include the Iterated Prisoner's Dilemma (Rapoport & Chammah, 1965), where two agents repeatedly decide whether to cooperate or defect, and the Coin Game (Lerer & Peysakhovich, 2018), in which two agents navigate a $3 \times 3$ grid to collect coins that appear in red or blue. Agents receive rewards for collecting any coin but incur penalties when their opponent collects a coin of their own color, creating tension between selfish coin-collecting and the Pareto-optimal "color-aligned" strategy. Another example is the Negotiation Game (Cao et al., 2018), where two agents simultaneously propose how to divide valuable items, each striving to secure items they highly value while risking worse outcomes if both demand large shares of the same limited resources. Furthermore, Diplomacy (Paquette), an adaptation of the classic Diplomacy board game tailored for multi-agent research, requires players to negotiate alliances, coordinate actions, and balance cooperative and adversarial incentives to expand territorial control without overextending themselves.

**Strategic behavior and climate change.**     Game theory has long studied the collective behavior of self-interested agents, e.g., the tragedy of the commons (Hardin, 1968), negotiation and agreements of agents with conflicting and common goals (Schelling, 1980). Certain works have analyzed international negotiations on climate collaboration and agreements regarding economic activity and climate efforts, e.g., imposing tariffs on countries that do not mitigate sufficiently. Experimental research highlights the collective action dynamics of climate cooperation. Uncertainty about the likelihood of catastrophic scenarios (Barrett & Dannenberg, 2012; Milinski et al., 2008) and interagent inequality (Dannenberg et al., 2015) has been found to hinder cooperation; however, interagent communication (Loschel et al., 2011) and early mitigation commitments facilitates it (Dannenberg et al., 2015). Climate negotiations have been studied using mathematical games, e.g., coordination games or prisoner's dilemmas (DeCanio & Fremstad, 2013). However, the reliability of such simplified models for real-world policy has been called into question. In particular, these games lack (i) a multilateral, rather than bilateral, setting, (ii) strategic behavior from agents with multiple, possibly conflicting, goals, (iii) evolving climate dynamics and changing agent behavior that lead to non-equilibrium outcomes, and (iv) heterogeneity among agents (Madani, 2013).

Subsequent work has gone beyond equilibrium analysis by modeling climate negotiations as a bargaining game in which agents learn, albeit in a highly simplified manner (Smead et al., 2014). Climate scenarios could be studied through the emergence of climate mitigation from these games with learning, in which regions can cooperate or compete (Greeven et al., 2016). Furthermore, other work has studied the difficulty of long-term climate collaboration (Carney, 2015), as well as potential mechanisms for overcoming associated issues (Nordhaus, 2015).

**Multi-agent reinforcement learning (MARL)**     MARL has emerged in recent years as an attractive framework that studies how to train utility-maximizing agents that may communicate, cooperate, or compete. This is a rich area of research that intersects machine learning with game theory, economics, and other domains (Shoham & Leyton-Brown, 2008). Games can be classified as cooperative, competitive, or a mixture of both. In fully cooperative games, agents learn to work together, e.g., to lower the carbon power consumption of heating, ventilation and air conditioning (HVAC) systems (Mai et al., 2024; Hanumaiah & Genc, 2021; Yu et al., 2020), or in the game of Hanabi (Yu et al., 2021). On the other hand, in a competitive game, agents may need to find strategies to defeat opponents, e.g., in Diplomacy (Paquette et al., 2019) and Go (Schrittwieser et al., 2020). However, many games are neither purely competitive nor purely cooperative (Duque et al., 2024). These are called mixed-motive games since incentives of agents are partly misaligned. A common and extensively studied example is public goods games (PGG) which describes the social dilemma between collaboration (contributing to common pot) which is the Pareto optimal outcome, and the free riding (keeping the resource for oneself) (Anderson et al., 1998; Santos et al., 2008; Orzan et al., 2024). In the context of climate change, it has also been studied studied (Tavoni et al., 2011).

Beyond abstract games, MARL has been increasingly applied to real-world scenarios where cooperation and competition are intertwined. For instance, Wang et al.(Wang et al., 2021) applied MARL to large-scale traffic signal control, balancing individual

intersection throughput with global traffic flow. In energy systems, May et al.(May & Huang, 2023) used MARL to model and optimize peer-to-peer prosumer markets, where agents must trade off self-interest and collective grid stability. Similarly, Hou et al. (Hou et al., 2025) introduce InvestESG, a MARL benchmark where agents need to navigate the trade-off between short-term profit goals and long-term climate resilience by strategically investing in mitigation, greenwashing, and adaptation measures.

Recent work has explored the link between MARL and negotiation (Cao et al., 2018), as well as cooperation in social dilemmas and collaboration on climate change (Jaques et al., 2019; Chelarescu, 2021; Le Gléau et al., 2022). As such, MARL is an attractive framework to analyze climate outcomes which takes strategic behavior into account. However, previous work has largely considered highly stylized environments and has not yet been applied to rich calibrated climate-economic simulations; our work fills this gap.

# B    Parameters and variables

Tables 2, 3, 4, 5 and 7 list all (calibrated) parameters and variables.

*Table 2.* **World-state variables.** Global type variables correspond to the entire world, whereas regional type variables correspond to each region. Endogenous variables are those which are affected by the agent actions, whereas exogenous variables are those that are predetermined and not affected by agent actions. Note that the values of endogenous variables can vary across steps in a predetermined manner. **Notation: indices are separated from subscripts referring to a name by semicolons (;). For instance, the parameter $\theta_1$ varies in time $t$ and by region $i$, which is denoted as $\theta_{1;i,t}$.**

| Variable | Type | Symbol | Description |
|---|---|---|---|
| Carbon Mass | Global, endogenous | $M_t, [M_t^{AT}, M_t^{UP}, M_t^{LO}]$ | A three-dimensional vector that indicates the average carbon accumulation in the atmosphere, upper oceans, and lower oceans. |
| Temperature | Global, endogenous | $T_t, [T_t^{AT}, T_t^{LO}]$ | A two-dimensional vector that indicates the average temperature of the atmosphere and the lower ocean. |
| Population | Regional, exogenous | $L_{i,t}$ | Population and the labor in a region. |
| Technology | Regional, exogenous | $A_{i,t}$ | Technology factor in the production function of a region. |
| Capital | Regional, endogenous | $K_{i,t}$ | Total capital accumulated by a region. |
| Carbon intensity of economic activity | Regional, exogenous | $\sigma_{i,t}$ | A scalar coefficient that gives the emissions resulting from economic production. |
| Balance of trade | Regional, endogenous | $D_{i,t}$ | Surplus or deficit from international trade activities. |
| Cost of mitigation efforts | Global, endogenous | $\theta_{1;i,t}$ | An estimate of the cost of mitigation efforts. |
| Emission due to land use | Regional, exogenous | $E_t^{\text{Land}}$ | Carbon emission for land use in a specific region. |

*Table 3.* **Agent-action variables.**

| Variable | Symbol | Description |
|---|---|---|
| Savings rate | $s_{i,t}$ | The fraction of output production to be invested in capital. |
| Mitigation rate | $\mu_{i,t}$ | The fraction of mitigation efforts by a region. |
| Import tariffs | $\tau_{i,j,t}$ | The fraction of imports that are converted to tariff revenue. |
| Export limits | $p_{i,t}^x$ | The fraction of domestic production that regions are willing to export. |
| Import bids | $b_{i,j,t}$ | The amount of production each region is willing to import from other regions. |

# C    The Activity Component: Climate, Economics, Trade, and Tariffs

## C.1    Climate and Economic Dynamics

We now describe the RICE-N dynamics developed from DICE and RICE models by (Nordhaus, 2018; Kellett et al., 2019) that govern the evolution of the world state from time $t$ to $t + 1$ for the different regions. Note that variables without an agent index are global quantities.

Table 4. **Agent-specific constants.**

| Variable | Symbol | Description |
|---|---|---|
| Initial population | $L_{0;i}$ | The initial population for a specific region. |
| Population convergence target | $L_{a;i}$ | The estimated convergence population for a specific region. |
| Population convergence rate | $l_{g;i}$ | How fast the current population converges. |
| Initial capital | $K_{0;i}$ | The initial capital for a specific region. |
| Initial carbon intensity | $\sigma_{0;i}$ | The initial carbon intensity for a specific region. |
| Carbon intensity parameters | $g_{\sigma;i}$ and $\delta_{\sigma;i}$ | The decay speed of the carbon intensity. |
| Initial technology factor | $A_{0;i}$ | The initial carbon technology factor for a specific region. |
| Technology factor parameter | $g_{i,A}$ and $\delta_{i,A}$ | The update pattern of the technology factor. |
| Initial land use emission | $E_{L0;i}$ | The initial land use emission for a specific region. |
| Land use emission parameter | $\delta_{EL;i}$ | The depreciation rate for the land use emission in a specific region. |

Table 5. **Global constants.**

| Variable | Symbol | Description |
|---|---|---|
| Capital elasticity of production | $\gamma$ | The contribution from capital and population to the economy. |
| Armington substitution parameter | $\lambda$ | How substitutable consumption goods from different regions are. |
| Long term welfare discount rate | $\rho$ | How much short-term welfare is weighted versus long-term welfare. |
| capital depreciation rate | $\Phi_K$ | The capital depreciation rate. |
| Backstop technology | $p_b$ | Price of a backstop technology that can remove carbon dioxide from the atmosphere. |
| Backstop technology parameter | $\delta_{pb}$ | The decay speed of the cost of backstop technology. |
| Mitigation efficiency parameter | $\theta_2$ | The efficiency loss component of mitigation |
| Domestic share parameter | $\psi^{dom}$ | The relative preference for domestic goods |
| Foreign share parameter | $\psi^{for}$ | The relative preference for foreign goods |

**Carbon mass.** The total carbon mass in the climate system is given by:

$$M_{t+1} = \Phi_M M_t + B_M \sum_i E_{i,t}, \tag{6}$$

$$E_{i,t} = E_t^{\text{Land}} + \sigma_{i,t}(1 - \mu_{i,t})Y_{i,t}, \tag{7}$$

$$M_t \doteq \begin{bmatrix} M_t^{\text{AT}} & M_t^{\text{UP}} & M_t^{\text{LO}} \end{bmatrix}^\top \in \mathbb{R}^3, \tag{8}$$

$$\Phi_M \doteq \begin{bmatrix} \zeta_{11} & \zeta_{12} & 0 \\ \zeta_{21} & \zeta_{22} & \zeta_{23} \\ 0 & \zeta_{32} & \zeta_{33} \end{bmatrix}, \tag{9}$$

$$B_M \doteq \begin{bmatrix} \xi_2 \\ 0 \\ 0 \end{bmatrix}. \tag{10}$$

This describes a three-reservoir model of the global carbon cycle, in which $M_{AT}$ describes the average mass of carbon in the atmosphere, $M_{UP}$ is the average mass of carbon in the upper ocean, and $M_{LO}$ the average mass of carbon in the deep or lower ocean, see Figure 5. $\Phi_M$ is the Markov transition matrix describing how carbon transfer between different reservoirs. $B_M$ describes how the weight of carbon emission affects the carbon accumulation in the reservoirs.

**Global temperature.** Ultimately, increasing carbon mass leads to rising temperatures:

$$T_{t+1} = \Phi_T T_t + B_T F_t, \tag{11}$$

$$T_t \doteq \begin{bmatrix} T_t^{\text{AT}} & T_t^{\text{LO}} \end{bmatrix}^\top \in \mathbb{R}^2, \tag{12}$$

$$F_t = F_{2\times} \log_2\left(\frac{M_t^{\text{AT}}}{M^{\text{AT},1750}}\right), \tag{13}$$

$$\Phi_T \doteq \begin{bmatrix} \phi_{11} & \phi_{12} \\ \phi_{21} & \phi_{22} \end{bmatrix}, \tag{14}$$

$$B_T \doteq \begin{bmatrix} \xi_1 \\ 0 \end{bmatrix}. \tag{15}$$

Similar to the carbon mass dynamic, there are two layers in the energy balance model, see Figure 4. $T_{AT}$ is the combined average temperature in atmosphere, land surface, and upper ocean (simply referred to as the "atmospheric layer" hereafter). $T_{LO}$ is the temperature in the lower ocean. $\Phi_T$ is the Markov transition matrix describing how heat transfers between different layers. $B_T$ describes how carbon mass contributes to the temperature increases.

**Output production.** The production in a region is given by the *total factor productivity (TFP)* (Comin, 2010) formula:

$$Y_{i,t} = A_{i,t} K_{i,t}^{\gamma} L_{i,t}^{1-\gamma}. \tag{41}$$

Production depends on three factors: total factor productivity ("technology") $A_t$, capital $K_t$, and labor $L_t$. This production function is common in the economic literature and used in the DICE/RICE models. The capital elasticity $\gamma \in [0,1]$ explains the different levels of contribution of capital and labor.

**Population.** The number of people in a region, denoted $L_t$ grows as:

$$L_{i,t+1} = L_{i,t} \left( \frac{1 + L_{a;i}}{1 + L_{i,t}} \right)^{l_{g;i}}. \tag{39}$$

There are two parameters $L_{a;i}$ and $l_{g;i}$. $L_{a;i}$ represents the convergence population of region $i$ and $l_{g;i}$ shows how fast the population $L_{i,t}$ converge to $L_{a;i}$. Please refer to the Appendix H for a more detailed analysis and the calibration procedure.

**Level of technology.** The technology factor $A_t$ describes how efficient production is, i.e., how many units of output a region achieves given fixed capital and labor:

$$A_{i,t+1} = (e^{\eta} + g_{A;i} e^{-\delta_{A;i} \Delta(t-1)}) A_{i,t}. \tag{40}$$

Here, $\eta$ represents the long-term growth of economics which is usually larger than 0, $g_A$ represents the short-term part of economics growth, and $\delta_A$ represents the speed of decay of short-term growth factor. $\Delta$ is the time difference between steps. We use $\eta = 0.33\%$ as in (Nordhaus, 2018).

**Capital.** The amount of capital evolves as:

$$\Phi_K \doteq (1 - \delta_K)^{\Delta}, \tag{16}$$

$$K_{i,t+1} = \Phi_{K,i} K_{i,t} + \Delta \left( 1 - a_1 T_t^{\text{AT}} - a_2 \left( T_t^{\text{AT}} \right)^2 \right) \tag{17}$$

$$\times \left( 1 - \theta_{1;i,t} \mu_{i,t}^{\theta_2} \right) Y_{i,t} s_{i,t}. \tag{18}$$

The evolution of the capital comes from two parts. The first part is capital inherited from the previous period with depreciation. In the second part, $s_t$ is a control variable which represents the investment/savings rate (as a fraction of production). That is, as a base amount, the economy invests/saves a total of $Y_{i,t} s_{i,t}$ which yields new capital. This base amount is further modified by 2 multipliers: the damage function and mitigation/abatement costs, which are discussed below.

**Damage function.** The climate damage function represents the economic damage due to climate change, e.g., increases in the atmosphere temperature $T_t^{\text{AT}}$. That is, in Equation 37, the fraction of new capital is modified by the damage function

$$1 - a_1 T_t^{\text{AT}} - a_2 \left( T_t^{\text{AT}} \right)^2, \tag{19}$$

following (Nordhaus, 2015). That is, higher temperatures lead to less new capital. Similarly, $1 - \theta_{1;i,t} \mu_t^{\theta_2}$ is the fraction of new capital after taking into account carbon emission mitigation. Mitigating carbon emissions more (higher $\mu_t$) means (dirty) production needs to be lowered, hence yields less new capital.

**Mitigation (abatement) cost.** Following (Kellett et al., 2019), for a mitigation rate $\mu_{i,t}$, the mitigation cost is

$$\theta_{1;i,t} \mu_{i,t}^{\theta_2} Y_{i,t} s_{i,t}, \tag{20}$$

where $\theta_{1;i,t}$ is given by Equation 38. This represents the loss in capital growth due to a fraction of production being used for mitigation.

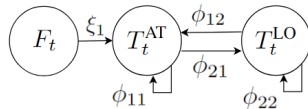

Figure 4. **The two-reservoir temperature model.**

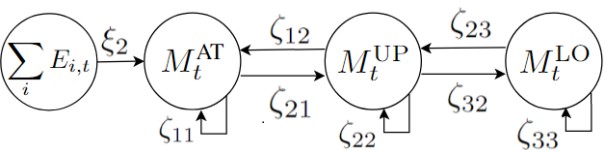

Figure 5. **The three-reservoir carbon mass model.**

**Carbon intensity of economic activity.** A critical part of the model is the interaction between the climate and economic parts. Specifically, the RICE model describes how production leads to carbon emissions:

$$E_t^{\text{Land}} = E_{L0} \cdot (1 - \delta_{EL})^{t-1}, \tag{21}$$

$$E_{i,t} = E_t^{\text{Land}} + \sigma_{i,t} A_{i,t} (1 - \mu_{i,t}) Y_{i,t} \tag{22}$$

$$\sigma_{i,t+1} = \sigma_{i,t} e^{-g_{\sigma;i}(1-\delta_{\sigma;i})^{\Delta(t-1)} \Delta}. \tag{23}$$

Here $E_t^{\text{land}}$ is the carbon emission due to (changes in) land use, $E_{L0}$ is the carbon emission in the base year, and $\delta_{EL}$ is the speed of decrease of changes in land use. The rates $0 < \delta_{EL} < 1, 0 < \delta_{L0} < 1$ are free parameters. Due to a lack of data, $E_t^{\text{land}}$ is set to be the same for each region.

$E_{i,t}$ is the total carbon emission, $E_t^{\text{land}}$ is emission from natural sources, while $E_{i,t} - E_t^{\text{land}}$ is emission caused by economic activity. $\sigma_{i,t} A_{i,t}$ is the effective carbon intensity of economic activity: a higher technology factor lead to higher emissions, but can be modulated by lower $\sigma$ (which can be thought of as the degree of "clean" production). $\mu_{i,t} \in [0, 1]$ is a control variable called the abatement (ratio), which represents the proportion of the economics contributing to reducing carbon emission. Furthermore, we have 2 parameters $g_\sigma$ and $\delta_\sigma$ that are fitted to data. $g_\sigma$ is the rate of decrease in carbon emissions.

### C.2 Trade

We now describe the international trade dynamics and the resulting regional consumption and utilities. Regions trade by exporting their own consumption goods and importing other regions' consumption goods at a fixed unit price[4].

**Agent actions.** Each region $i$ at time $t$ must first specify a desired basket of consumption goods $\boldsymbol{b_{i,t}} = [b_{i,1,t}, ..., b_{i,k,t}]$ that they are willing to import from the other regions. These desired imports form a matrix of *bids* $B_t$ such that the import bid by region $i$ for goods from $j$ at time $t$ is $b_{i,j,t} \geq 0$, i.e., the amount of goods region $i$ is willing to import from region $j$ at time $t$ is $b_{i,j,t}$.

Regions also set an upper bound $p_{i,t}^x \in [0, 1]$ on the proportion of their own consumption goods that they are willing to export.

**Tariffs.** Regions can also choose to impose import tariffs on other regions. We denote an import tariff imposed by region $i$ on a region $j$ by $\tau_{i,j,t} \in [0, 1]$. If region $i$ imposes an import tariff $\tau_{i,j,t} \in [0, 1]$ on region $j$, region $i$ consumes

$$C_{i,j,t} = x_{i,j,t} (1 - \tau_{i,j,t}), \tag{24}$$

and $\tau_{i,j,t} x_{i,j,t}$ is added to a reserve fund specific to that region.

**Consumption.** Consumption of domestic goods $C_{i,i,t}$ is determined according to gross output, the savings rate and exports:

$$C_{i,i,t} = (1 - s_{i,t}) Q_{i,t} - \sum_{j \neq i} x_{j,i,t}. \tag{25}$$

---

[4]More generally, prices should be dynamic, but the current implementation does not support this.

The aggregated consumption $C_{i,t}$ at time $t$ for region $i$ is given by the Armington elasticity model (Lessmann et al., 2009)) as follows:

$$C_{i,t} = \left( \psi^{dom}(C_{i,i,t})^{\lambda} + \sum_{j \neq i} \psi^{for}(C_{i,j,t})^{\lambda} \right)^{\frac{1}{\lambda}}. \tag{26}$$

### C.3 Negotiations

Here, we outline the extra state representations and actions that are introduced when either of our two negotiation protocols is active in the simulation. These variables are used in the negotiation phases, which are detailed in Section 4, on top of the already available variables in Tables 2, 3, 4, 5 and 7.

Table 6 presents the agent actions required by the Bilateral Negotiation and Basic Club protocols. These actions are additionally incorporated into the observation space of the agents at each timestep as bilateral observations, meaning that an agent will observe its own negotiation action and the action of other agents when it is part of that negotiation. In addition, each agent observes an additional indicator that signals the current phase of the environment (proposal, evaluation, or no negotiation step). Lastly, agents also privately observe the outcome of their negotiation: their minimum mitigation rate or club mitigation rate $\mu_c$.

We note that other negotiation protocols may modify these action and observations spaces as needed.

*Table 6.* **Agent-action variables introduced when one of our negotiation protocols is enabled in RICE-N.** These variables are additionally added as bilateral observations to the observation space. In the Basic Club protocol however, the request action is excluded, and each agent proposes a single mitigation rate to all agents. The result of these negotiation actions sets an agent's minimum mitigation rate or club mitigation rate $\mu_c$.

| Variable | Symbol | Description |
|---|---|---|
| Proposed mitigation rate | $\hat{\mu}_{i,j,t} / \hat{\mu}_{i,t}$ | The minimum mitigation rate proposed by agent $i$ to agent $j$. In basic club, a single proposal is made by each agent to all other agents. |
| Requested mitigation rate | $\bar{\mu}_{i,j,t}$ | The minimum mitigation rate requested from agent $j$ by agent $i$. Only used in the Bilateral Negotiation protocol and not in Basic Club. |
| Proposal decisions | $e_{i,j,t}$ | The decision of agent $i$ on the proposal and request by agent $j$. |

## D RICE-N dynamics

At a high-level, Equations 35 and 36 capture climate dynamics (temperature and carbon mass), while Equations 37, 39, 40, and 41 capture economic dynamics. Finally, Equation 42 captures the carbon-intensity of production, providing a key link between the climate and economic sectors.

## E Computational Complexity

The computational complexity of our MARL approach is driven by the number of regions $N$ (i.e., agents). Since each agent's action space scales linearly with $n$, the total action space across all agents grows quadratically ($\mathcal{O}(n^2)$). However, the number of agents in our setting is naturally bounded by the number of countries on the planet. Currently, training 27 agents for 100 thousand episodes takes approximately 3 hours on a 30 CPU cluster. Future efforts will be directed at more efficient implementations using JAX-based acceleration and model parallelism to improve runtime, which will enable large-scale sensitivity analyses and experiments.

## F Creating a 27-Region Simulation

We feature $n = 27$ fictitious regions in our simulation. These are inspired by merging and splitting real-world countries, but are not exactly the same as real-world regions.

We used real data from the World Bank API (WorldBank, 2022), e.g., GDP, capital stock, population, and $CO_2$-quivalent ($CO_2$eq) emissions. Furthermore, the World Bank groups countries into regions, including Sub-Saharan Africa, South Asia, North America, the Middle East and North Africa, Latin America and the Caribbean, Europe and Central Asia, East Asia

---

**Algorithm 1** Activity Component (implemented by `Climate_and_economy_simulation_step()`) Note that we only list input state variables and omit model parameters.

---

**Require:** exogenous emissions, land emissions, intensity, production factor, labor, capital,
  previous global temperature, previous government balance
**Require:** actions : mitigation rates, saving rates, tariffs, export rate limit, desired imports
  **for** each region **do**
    mitigation cost $\leftarrow f$(intensity)         ▷ Equation 38
    damages $\leftarrow f$(previous global temperature)         ▷ Equation 19
    abatement cost $\leftarrow f$(mitigation rate, mitigation cost)         ▷ Equation 20

    production $\leftarrow f$(production factor, capital, labor)         ▷ Equation 41
    gross output $\leftarrow f$(damages, abatement cost, production)         ▷ Equation 41
    government balance $\leftarrow f$(interest rate, previous government balance)
    investment $\leftarrow f$(saving rate, gross output)         ▷ Using Equation 37

    scaled imports $\leftarrow f$(gross output, desired imports)         ▷ Equation 55
    debt ratio $\leftarrow f$(previous government balance)         ▷ Equation 56
    scaled imports $\leftarrow f$(scaled imports, debt ratio)         ▷ Equation 57
  **end for**
  **for** each region **do**
    max potential exports $\leftarrow f$(gross output, investment, export rate limit)         ▷ Equation 58
    Scaled imports $\leftarrow f$(scaled imports, max potential exports)         ▷ Equation 59
  **end for**
  **for** each region **do**
    tariff-ed imports, tariff revenue $\leftarrow f$(scaled imports, tariffs)         ▷ Equation 24
    domestic consumption $\leftarrow f$(savings, gross output, scaled imports)         ▷ Equation 25
    aggregate consumption $\leftarrow f$(domestic consumption, tariff-ed imports)         ▷ Equation 26
    utility $\leftarrow f$(labor, aggregate consumption)         ▷ Equation 2
    government balance $\leftarrow f$(imports, exports)         ▷ Equation 60
  **end for**
  temperature $\leftarrow f$(previous temperature, previous carbon mass, exogenous emissions)
  carbon mass $\leftarrow f$(previous carbon mass, intensity, mitigation rate, production, land emissions)
  **for** each region **do**
    capital $\leftarrow f$(capital, investment)         ▷ Equation 37
    labor $\leftarrow f$(labor)         ▷ Equation 39
    production factor $\leftarrow f$(capital)         ▷ Equation 40
    carbon intensity $\leftarrow f$(carbon intensity)         ▷ Equation 42
  **end for**

---

and Pacific. In each region, the different countries (or sub-regions) are classified into 4 income groups: high income, upper middle income, lower middle income, and low income.

**Merging regions.** We assume the GDP, capital stock, and population for the regions are additive. We also assume the gross $CO_2$eq emissions across the regions are additive. Thus, we have

$$K_m = \sum_i K_i, \tag{27}$$

$$L_m = \sum_i L_i, \tag{28}$$

$$Y_m = \sum_i Y_i, \quad \text{where} \quad Y_i := A_i K_i^\gamma L_i^{1-\gamma}, \tag{29}$$

$$A_m = \frac{Y_m}{K_m^\gamma L_m^{1-\gamma}}, \tag{30}$$

$$\sigma_m = \frac{\sum_i \sigma_i Y_i}{Y_m}. \tag{31}$$

Note that the production function is not scale-invariant:

$$Y_t = (A_t K_t)^\gamma (A_t L_t)^{1-\gamma} \tag{32}$$

$$c \cdot Y_t = (c \cdot A_t K_t)^\gamma (c \cdot A_t L_t)^{1-\gamma} \tag{33}$$

$$\neq (c \cdot A_t)(c \cdot K_t)^\gamma (c \cdot L_t)^{1-\gamma}, \quad \forall c > 0. \tag{34}$$

Hence, one cannot get the technology after merging multiple regions by simply adding the individual technology levels. Rather, the combined technology factor is imputed from the combined productions, labor, and capital.

$$T_{t+1} = \Phi_T T_t + B_T \left( F_{2\times} \log_2 \left( \frac{M_t^{\text{AT}}}{M^{\text{AT},1750}} \right) + F_t^{\text{EX}} \right), \tag{35}$$

$$M_{t+1} = \Phi_M M_t + B_M \left( \sum_i \sigma_{i,t}(1 - \mu_{i,t})Y_{i,t} + E_t^{\text{Land}} \right), \tag{36}$$

$$K_{i,t+1} = \Phi_{K,i} K_{i,t} + \Delta \left( 1 - a_1 T_t^{\text{AT}} - a_2 \left( T_t^{\text{AT}} \right)^2 \right) \left( 1 - \theta_{1;i,t}\mu_{i,t}^{\theta_2} \right) Y_{i,t} s_{i,t}, \tag{37}$$

$$\theta_{1;i,t} = \frac{p_b}{1000 \cdot \theta_2} (1 - \delta_{pb})^{t-1} \cdot \sigma_{i,t}, \tag{38}$$

$$L_{i,t+1} = L_{i,t} \left( \frac{1 + L_{a;i}}{1 + L_{i,t}} \right)^{l_{g;i}}, \tag{39}$$

$$A_{i,t+1} = (e^\eta + g_{A;i}e^{-\delta_{A;i}\Delta(t-1)})A_{i,t}, \tag{40}$$

$$Y_{i,t} = A_{i,t}K_{i,t}^\gamma L_{i,t}^{1-\gamma}, \tag{41}$$

$$\sigma_{i,t+1} = \sigma_{i,t}e^{-g_{\sigma;i}(1-\delta_{\sigma;i})^{\Delta(t-1)}\Delta}. \tag{42}$$

**Splitting large regions.** To avoid huge economies that dominate the fictitious world, we split large economies into pieces based on predetermined fractions $c_i$ and random sampled $A_i$:

$$\sum_i c_i = 1, \tag{43}$$

$$L_i = c_i L_m, \tag{44}$$

$$Y_i = c_i Y_m, \tag{45}$$

$$K_i = \frac{Y_i}{A_i L_i^{1-\gamma}}, \tag{46}$$

$$\sigma_i = \sigma_m. \tag{47}$$

## G   Welfloss

Welfloss, $w$ stands for loss of welfare due to imposed tariffs (Nordhaus, 2015). $w$ relies on $wl$, the unit of welfare loss per unit of tariff which Nordhaus calibrates to .4.

$$wl = .4 \tag{48}$$

$$w_{i,t} = 1 - wl \sum_j \frac{b_{i,j,t}}{Y_{i,t}} \tau_{i,j,t} \tag{49}$$

## H   Model Calibration

The structural parameters of the RICE-N simulation were calibrated to meet the following objectives:

1. Temperatures match the real data in different versions of RICE-N with 3 regions, 7 regions, 20 regions, 27 regions, and 189 regions, under 0% and 100% mitigation.

2. The optimistic-pessimistic temperature outcomes fit the projects of Shared Socioeconomic Pathways in the IPCC Sixth Assessment Report (Pörtner et al., 2022) ($2°C$ - $5°C$ increase in the year of 2100). Each region optimizes the target without negotiation and direct cooperation in the pessimistic case. In the optimistic case, regions negotiate with each other using the baseline bilateral negotiation protocol. Please also notice that in the extremely pessimistic case that regions ignore climate change at all and always choose 0% mitigation and 100% savings, the temperature leads to approximately $7°C$ increase in the year 2100.

The parameters that we estimated and the corresponding estimation methods are listed below:

- The dynamic parameters for total factor productivity $A$: $g_A$ and $\delta_A$.

- The capital $K$: for the regions whose capital data is not available, we use a KNN regressor (Buitinck et al., 2013) to estimate it.

- The dynamic parameters for population $L$: $l_g$; similarly, for the regions whose convergence population data is not available, we use a KNN regressor to estimate it.

- The initial carbon intensity $\sigma_0$: for the regions whose capital data is not available, we use a KNN regressor to estimate it.

- KNN regressor: Because all regions have GDP and population data, we use them as features. For each region that lacks emission data and capital data, we find the nearest 5 neighbors according to its GDP and population. We use the average of the 5 neighbors' emission data and capital data as the estimated values.

### H.1 Population dynamic calibration

Denoting $L_{\infty;i} := \lim_{t\to\infty} L_{i,t}$, in the limit $t \to \infty$ we have:

$$L_{\infty;i} = L_{\infty;i} \left( \frac{1 + L_{a,i}}{1 + L_{\infty;i}} \right)^{l_{g;i}}, \tag{50}$$

$$1 = \left( \frac{1 + L_{a;i}}{1 + L_\infty} \right)^{l_{g;i}}. \tag{51}$$

As long as $l_{g;i}$ is not zero, $L_{\infty;i} = L_{a;i}$. Thus, $L_{a;i}$ is the long-term population size and a free parameter that is fitted to data. Assuming $\{L_{i,t}\}_{t=1,2,\dots}$ is monotonically increasing or decreasing, the absolute value of $l_{g;i}$ represents how fast it converges to $L_{a;i}$. The closer $L_{i,t}$ is to monotonically increasing or monotonically decreasing in the real data, the easier it is to fit $l_{g;i}$ and $L_{a;i}$.

To fit the population parameters, we take logs on both sides of Equation 39:

$$\log L_{i,t+1} =$$
$$\log L_{i,t} + l_{g;i}(\log(1 + L_{a;i}) - \log(1 + L_{i,t+1})), \tag{52}$$

where $\log L_{i,t+1} - \log L_{i,t}$ and $\log(1 + L_{i,t})$ are given by the data. $\log(1 + L_{i,t})$ and $l_{g;i}$ can then be estimated by linear regression.

### H.2 Technology dynamic calibration

We estimate both $g_A$ and $\delta_A$ from the existing data $\{A_t\}_{i=1\cdots n}$ by solving a regression problem:

$$g^*_{a;i}, \delta^*_{a;i} = \underset{g_{a;i},\delta_{A,i}}{\arg\max} \mathcal{L}_{i,t} \tag{53}$$

$$\mathcal{L}_{i,t} = ||A_{i,t+1} - (\exp\eta + g_{A,i}\exp(-\delta_{A,i}\Delta(t-1)))A_{i,t}||^2. \tag{54}$$

This can be solved by numerical optimization algorithms, e.g., as provided in SciPy (Virtanen et al., 2020).

Because the emissions data from the World Bank API do not fit the form of the $\sigma$ dynamic as assumed by DICE2016, use the DICE2016 parameter values for $g_\sigma$ and $\delta_\sigma$.

*Table 7.* Calibrated parameters for 27 regions

| Region ID | $A_0$ | $K_0$ | $L_0$ | $L_a$ | $\delta_A$ | $g_A$ | $l_g$ | $\sigma_0$ |
|---|---|---|---|---|---|---|---|---|
| 1 | 1.872 | 0.239 | 476.878 | 669.594 | 0.139 | 0.122 | 0.034 | 0.456 |
| 2 | 8.405 | 3.304 | 68.395 | 93.497 | 0.188 | 0.103 | 0.058 | 0.529 |
| 3 | 3.558 | 0.109 | 64.122 | 135.074 | 0.161 | 0.127 | 0.026 | 0.816 |
| 4 | 1.927 | 1.424 | 284.699 | 465.308 | 0.244 | 0.134 | 0.024 | 1.221 |
| 5 | 8.111 | 0.268 | 28.141 | 23.574 | 0.163 | 0.106 | -0.057 | 0.290 |
| 6 | 4.217 | 3.184 | 548.754 | 560.054 | 0.170 | 0.095 | 0.080 | 0.302 |
| 7 | 2.491 | 0.044 | 46.489 | 59.988 | 0.058 | 0.049 | 0.037 | 0.420 |
| 8 | 2.525 | 1.080 | 69.194 | 100.016 | 0.346 | 0.079 | 0.029 | 1.010 |
| 9 | 2.460 | 0.184 | 513.737 | 1867.771 | 1.839 | 0.462 | 0.017 | 0.310 |
| 10 | 12.158 | 2.642 | 38.101 | 56.990 | 0.131 | 0.063 | 0.020 | 0.350 |
| 11 | 0.993 | 0.160 | 522.482 | 1830.325 | 0.086 | 0.065 | 0.019 | 0.235 |
| 12 | 5.000 | 2.289 | 165.293 | 230.191 | 0.183 | 0.071 | 0.027 | 0.419 |
| 13 | 29.854 | 2.020 | 165.751 | 216.927 | 0.088 | 0.075 | -0.002 | 0.254 |
| 14 | 23.315 | 3.039 | 109.395 | 143.172 | 0.088 | 0.075 | -0.002 | 0.254 |
| 15 | 29.854 | 0.687 | 56.355 | 73.755 | 0.088 | 0.075 | -0.002 | 0.254 |
| 16 | 10.922 | 0.606 | 705.465 | 532.497 | 0.096 | 0.168 | -0.016 | 0.781 |
| 17 | 9.634 | 0.608 | 465.607 | 351.448 | 0.096 | 0.168 | -0.016 | 0.781 |
| 18 | 8.621 | 0.453 | 239.858 | 181.049 | 0.096 | 0.168 | -0.016 | 0.781 |
| 19 | 3.190 | 0.129 | 690.002 | 723.513 | 0.054 | 0.068 | -0.013 | 0.949 |
| 20 | 2.034 | 0.381 | 455.401 | 477.518 | 0.054 | 0.068 | -0.013 | 0.949 |
| 21 | 13.220 | 16.295 | 502.410 | 445.861 | 0.252 | 0.074 | -0.033 | 0.170 |
| 22 | 3.190 | 0.044 | 234.601 | 245.994 | 0.054 | 0.068 | -0.013 | 0.949 |
| 23 | 6.387 | 1.094 | 317.880 | 287.533 | 0.194 | 0.237 | -0.053 | 0.840 |
| 24 | 2.481 | 0.090 | 94.484 | 102.997 | 0.203 | 0.201 | 0.037 | 1.665 |
| 25 | 10.853 | 17.554 | 222.891 | 168.351 | 0.005 | 0.000 | -0.012 | 0.285 |
| 26 | 4.135 | 1.002 | 103.294 | 87.418 | 0.158 | 0.123 | -0.063 | 0.601 |
| 27 | 2.716 | 1.034 | 573.818 | 681.210 | 0.097 | 0.101 | 0.043 | 0.638 |

# I  Trade constraints

To ensure that total imports and total exports match, three constraints are enforced on regions' trade flows.

1. For each region $i$, if the region's total desired imports from other regions exceed its own gross output, then the imports are scaled to sum up to the region's gross output. We enforce the constraint that $\sum_{i \neq j} b_{i,j,t} \leq Q_{i,t}$, which is to say that a region may not import more goods than its current gross output capacity. This constraint helps the agents avoid insurmountable debt, thereby stabilizing trade balances over the entire time period while also easing learning. If a region's desired imports exceed its production capacity, then its import bids are scaled down to size :

$$ b_{i,j,t} \leftarrow b_{i,j,t} \min \left\{ 1, \frac{Q_{i,t}}{\sum_{i \neq j} b_{i,j,t}} \right\}. \tag{55} $$

2. Regions are allowed to carry a (positive or negative) trade balance $D_{i,t}$. At the start of each new time step, each region's trade balance, positive or negative, accumulates interest at a fixed rate of 10%. Based on this balance, a region's debt-to-initial-capital ratio is determined and the imports are scaled according to this ratio:

$$ d_{i,t} = 10 \frac{D_{i,t}}{K_0}, \tag{56} $$

$$ b_{i,j,t} \leftarrow b_{i,j,t}(1 + d_{i,t}). \tag{57} $$

3. If other regions' total desired imports from region $i$ exceed region $i$'s upper bound on exports $x_{i,t}^{\max}$, then the bids for goods from region $i$ are scaled proportionally to $x_{i,t}^{\max}$. Otherwise, each region receives its full import bid from region $i$. In other

words, region $i$ cannot export more goods at time $t$ than it could consume at time $t$, so other regions will import less from region $i$.

$$x_{i,t}^{\max} = \min(p_{i,t}^x Q_{i,t}, Q_{i,t} - I_{i,t}), \tag{58}$$

$$x_{i,j,t} = b_{i,j,t} \min \left\{ 1, \frac{x_{i,t}^{\max}}{\sum_{j \neq i} b_{i,j,t}} \right\}. \tag{59}$$

After all constraints have been applied, the trade balance for the next period is calculated:

$$D_{i,t+1} = D_{i,t} + \Delta \left( \sum_{j \neq i} x_{j,i,t} - \sum_{j \neq i} x_{i,j,t} \right). \tag{60}$$

## J Legal Framework

Tariff enforced mechanisms, such as Basic Club, must comply with the World Trade Organization's (WTO) General Agreement on Tariff and Trade (GATT); specifically, the "most favored nation" clause which requires that tariffs be non-discriminatory. At face value, Basic Club would appear to violate the clause; however, exceptions are made in the following circumstances:

- The agreement promotes one of the GATT article XX (g) objectives; namely, "relating to the conservation of exhaustible natural resources."

- The agreement should contribute to the objective.

- The agreement should not discriminate between countries. If it appears to, then its discrimination must be on the grounds justifies the rationale.

This legal framework has precedent since the 1998 WTO Appellate Body Report "United States - Import Prohibition of Certain Shrimp and Shrimp Productions" (Shaffer, 1999). Basic Club inherits this legal framework with respect to GATT compliance. Furthermore, tariffs can be WTO compliant if they correct existing trade imbalances, as is the case with carbon leaking regions which have a competitive advantage, or are used as a punitive measure against misconduct (Pihl, 2020; Mavroidis & de Melo, 2015).

## K Sensitivity Analysis

We carry out a sensitivity analysis to test the robustness of the results under different parameter settings. Since the space of possible configurations is large, we perform a sensitivity analysis over a subset of economically relevant parameters, namely the discount factor, welfare loss weight, consumption substitution rate and relative preference for domestic goods. Figure Appendix J shows the percentage change in outcome variables of interest across different scenarios when critical model parameters are perturbed by a multiplication factor ranging from 0.96 to 1.04. The maximum percentage change is 3.16% while the mean is $-0.22\%$ and the medium is $-0.36\%$. We thus conclude that the dynamics are stable, corresponding to changes in critical model parameters.

## L Mitigation Distribution

To explore the range of strategies that emerge under different negotiation protocols, we analyze the distribution of final mitigation rates of each agent. This clarifies for each negotiation protocol what proportion of agents are ambitious mitigators, free-riders, or low-effort mitigators. Results are visible in Figure 9.

## M Detailed Outcome and Fairness Times Series

Figure 7 and Figure 8, provide the timeseries and equity of various variables over relevant scenarios. The bump in carbon emissions within the first time steps is a result of the constraint that regions cannot abruptly change their mitigation rate, but only adapt it stepwise, leading to a slow ramp-up of mitigation at the start of the rollout. Even with the maximum mitigation rate, a base emission level remains, as land emissions are assumed to be non-reducible.

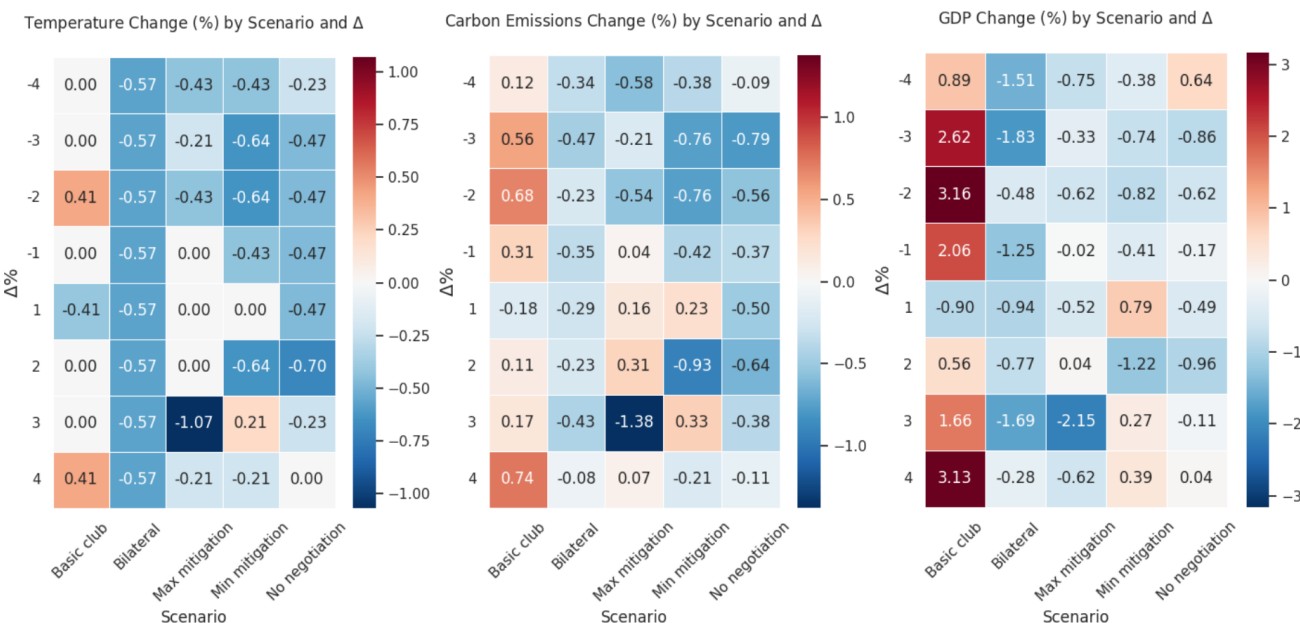

*Figure 6.* Sensitivity analysis: heatmap showing the percentage change in variables of interest (including Temperature, Carbon Emissions, GDP) across different scenarios (x-axis) when critical model parameters (including the discount factor, welfare loss weight, consumption substitution rate and relative preference for domestic goods) are perturbed by a multiplication factor of $(1 + \Delta)$. The parameter $\Delta$ varies from $-0.04$ to $0.04$ (y-axis). For example, looking at the bottom left corner of the Temperature Change heatmap, a $4\%$ decrease in the model parameters leads to an increase in final temperature of $0.41\%$. Overall, the variables of interest are rather insensitive to changes in the model parameter values. For reference, the global temperature anomaly increases by over $200\%$ from 2015 to 2115 in the Maximum mitigation scenario.

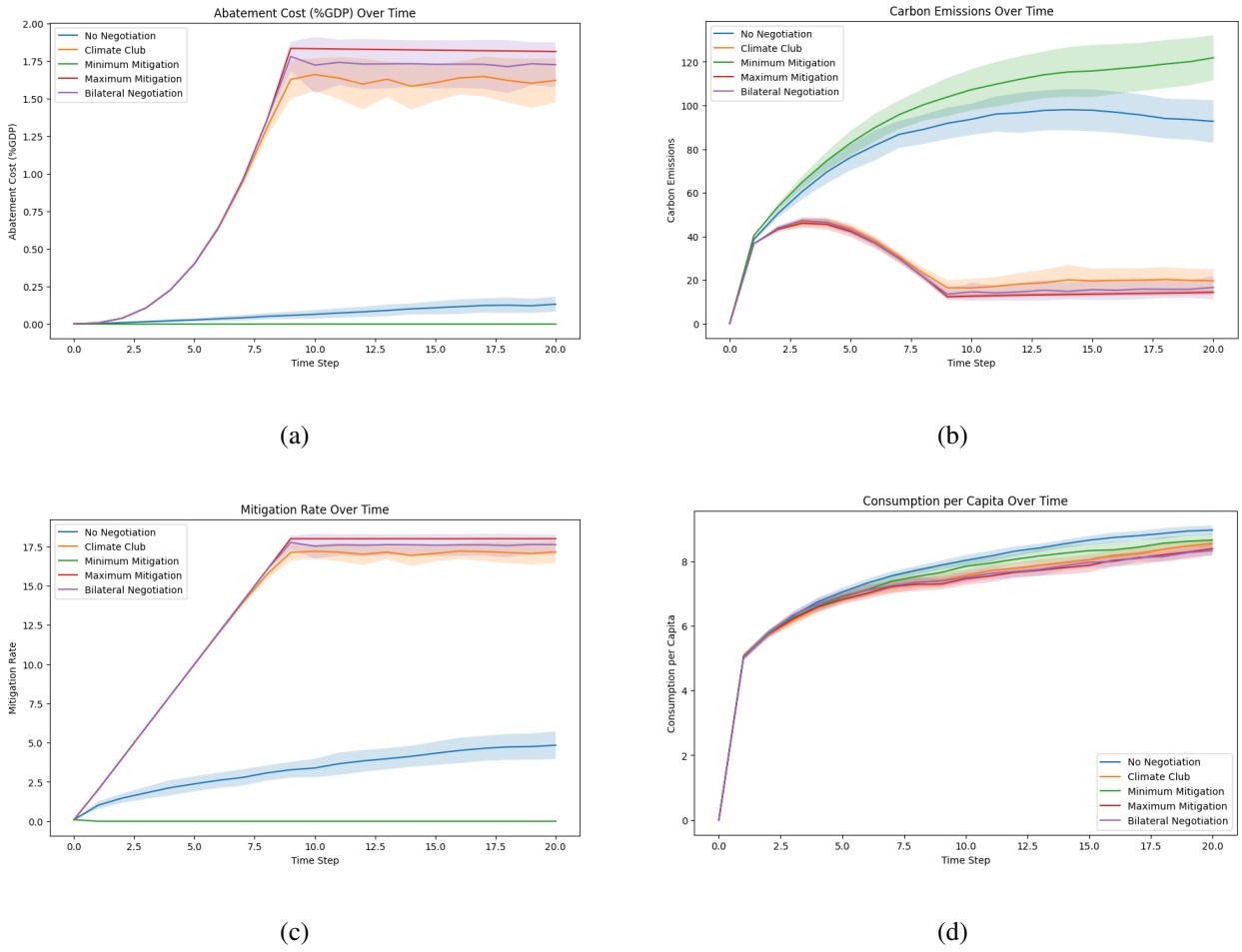

(a)

(b)

(c)

(d)

*Figure 7.* Time series of key variables across various scenarios.

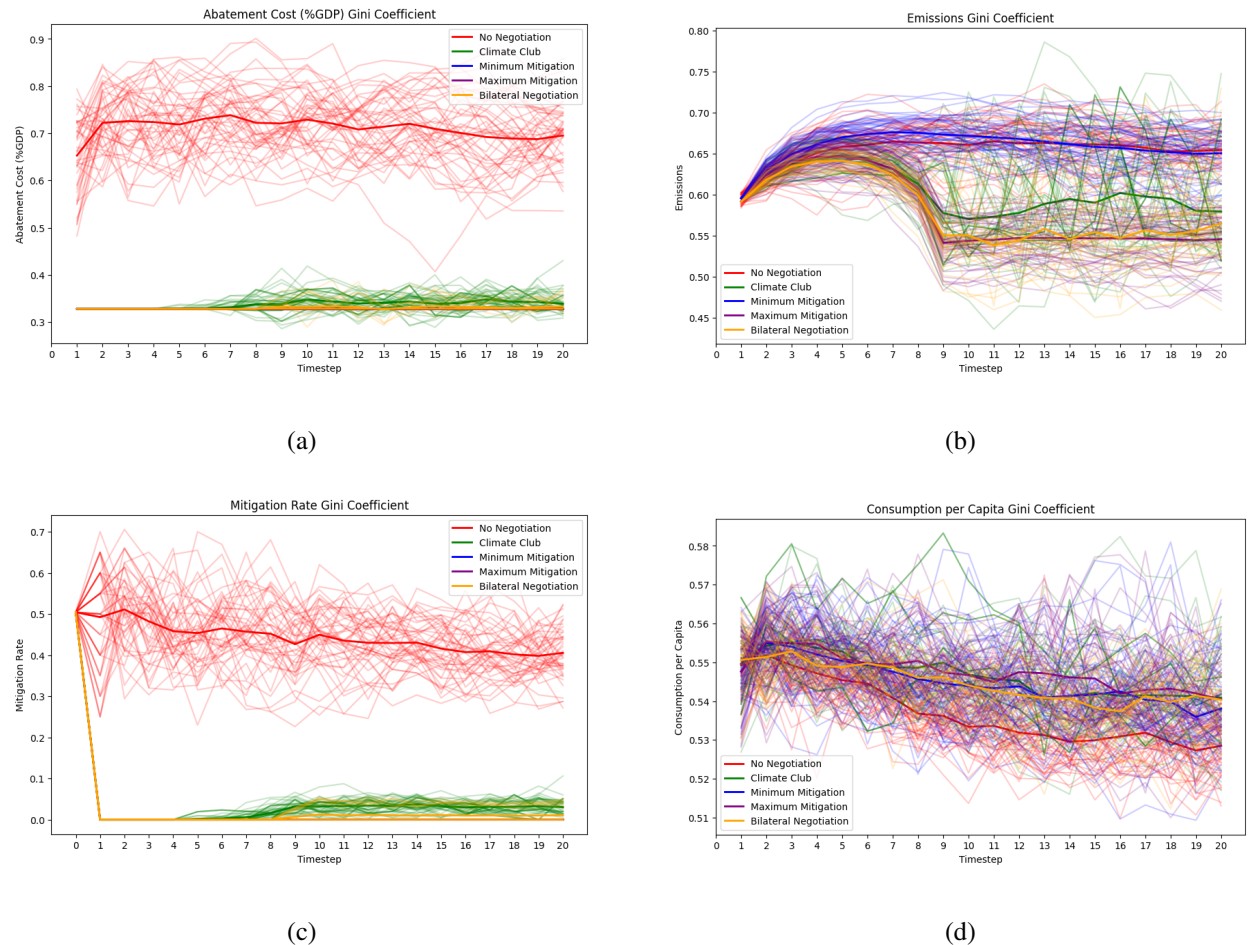

*Figure 8.* Equity of key variables across various scenarios.

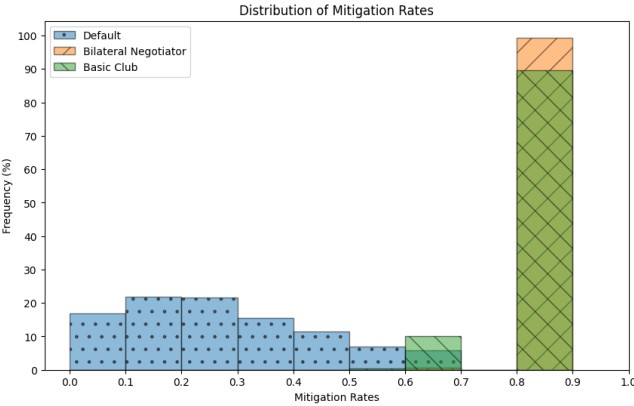

*Figure 9.* We compare the distribution of final mitigation rates across 50 seeds. Under the default, no negotiation, most regions either free ride or reduce ≤ 30% of their emissions. Under the Basic Club and Bilateral Negotiation, the majority of agents reduce 80% of their emissions.

