# OpenReview forum: "AI for Global Climate Cooperation: Modeling Global Climate Negotiations, Agreements, and Long-Term Cooperation in RICE-N"
_ICML.cc/2025/Conference — ICML 2025 poster_

### Official Review · Reviewer_ntBn · 2025-03-07

**Overall Recommendation:** 4

**Summary:**

## update after rebuttal

The paper proposed a multi-agent reinforcement learning (MARL) approach to model global climate negotiations, agreements, and long-term cooperation. Two novel negotiation protocols were proposed: Bilateral Negotiation and Basic Club. The proposed approach and protocols take into account many realistic components such as climate dynamics and economic dynamics. The simulated results compared the two negotiation protocols and a baseline without negotiation. The comparison shows better results with negotiation.

The rebuttal responses helped clarification.

**Claims And Evidence:**

The claim, the two negotiation protocols are better than the baseline without negotiation, is well supported by the experimental results.

**Essential References Not Discussed:**

I think the references in the Related Works (both main paper and appendix) are comprehensive.

**Experimental Designs Or Analyses:**

The experimental design in Section 6 and result analyses (mostly in Table 1) are reasonable.

**Methods And Evaluation Criteria:**

1. The proposed multi-agent reinforcement learning method and the two negotiation protocols are novel and make sense.
2. The evaluation criteria include both climate-economic and Gini indexes, which make sense and are comprehensive.

**Other Comments Or Suggestions:**

Figures 6-15 are not explained.

**Other Strengths And Weaknesses:**

W1: Most results are global macro level. Analysis of actions and goals of individual countries/regions are missing. The simulation should be reasonable at both global macro level and individual micro-level.

**Questions For Authors:**

Q1. Could region level analysis be provided? For example, what are a region’s actions overtime and how they are different with the two negotiation protocols and without negotiation. Are there any patterns in the actions for the 27 regions studied. Could their actions be categorized into a few common groups?

Q2. Can you summarize possible usages of the proposed approach for different types of users: ML researchers, climate scientists, government, international organizations such as IPCC?

**Relation To Broader Scientific Literature:**

The work extends existing climate-economic policy tradeoff modeling work including integrated assessment models (IAMs) and Regional Integrated Model of Climate and Economy (RICE) by capturing the strategic behavior in climate negotiations.

**Theoretical Claims:**

The work is a game-theoretic problem.  The paper itself does not have proof.

---

> ### Author Rebuttal · Authors · 2025-04-01
>
> We thank the reviewer for their thoughtful feedback and for pointing out the positive aspects of our submission, such as the novelty of the protocols, the realism of the components, the empirical support for our claims and the overall quality of the evaluation criteria, experimental design and results analysis. We hope our response clarifies any outstanding concerns.
>
> > Could region level analysis be provided? For example, what are a region’s actions overtime and how they are different with the two negotiation protocols and without negotiation. Are there any patterns in the actions for the 27 regions studied. Could their actions be categorized into a few common groups?
>
> Thank you for this suggestion. Under Basic Club, all regions tend to increase their mitigation rate as much as constraints allow until the maximum mitigation rate is reached. Without negotiation acting as a coordination mechanism, regions tend to fall into different subgroups. There are ambitious mitigators, reducing 40-60% of emissions by the end of the run, moderate mitigators reducing 20-30%, and free riders reducing 0-10%. In the camera-ready version, we will include a detailed quantitative and qualitative analysis of these phenomena.
>
> > Can you summarize possible usages of the proposed approach for different types of users: ML researchers, climate scientists, government, international organizations such as IPCC?
>
> ML researchers can make use of the modularity of the RL component to evaluate how different RL algorithms perform in a real world-calibrated environment. Climate scientists can use it to compare different climate modules and damage functions. Governments can use it to evaluate the robustness of a given climate-economic policy to strategic behavior and estimate the consequences on international trade. International organizations, such as the OECD or WTO could use the tool to analyze the economic trade-offs of different climate policies and negotiation strategies. We will add these potential uses to the text, while making sure to surface the limitations.
>
> > Figures 6-15 are not explained.
>
> Thank you for pointing this out. These results are additional visualizations to support Section 6. In the camera-ready, we will expand the captions for each of the images such that they are sufficiently informative to understand the figures, describing the various scenarios, as well as the variables under study, and what can be concluded from the plots. We will also point to them more strongly in the main text.

---

### Official Review · Reviewer_j6L5 · 2025-03-14

**Overall Recommendation:** 3

**Summary:**

The paper adds reinforcement learning to a multi-agent Nordhaus RICE model.

It allows two communication protocols between agents, whereby agents can make binding commitments to curtail their greenhouse gas emissions.

Simulation allows these two protocols to be compared to the 'no negotiation' (Nordhaus RICE) baseline.  Relative to that baseline, the negotiation protocols allow higher welfare outcomes, better controlling the climate's evolution.

The paper claims that the underlying code's modularity allows climate dynamics and economic loss functions to be easily adjusted.

## update after rebuttal

Score maintained for existing manuscript.  A considerably revised manuscript (as discussed in the rebuttal) could receive a higher score.

I am concerned by the statement that "no single equilibrium concept applies in all cases", as it raises the question of whether the output corresponds to any intelligible or benchmark description of play.  I suspect that nation states are fumbling their way forward here, rather than playing anything that could be clearly rationalized as a form of Nash equilibrium, so this may be fine - but needs to be argued.

Again, the lack of realism in the modeled mechanisms weakens the paper.

**Claims And Evidence:**

The headlines claims are evidenced by three figures of plots and a table in the body of the paper.

Overall, I did not find the claims surprising: stronger protocols for binding commitments allow agents to better overcome the 'cost of anarchy' associated with playing Nash equilibria rather than the social planner's first best outcome.

**Essential References Not Discussed:**

N/A

**Experimental Designs Or Analyses:**

Have not checked anything in detail.

**Methods And Evaluation Criteria:**

The various runs are compared on the basis of temperature deviations, GDP impact and other standard criteria.

Nordhaus' RICE model is a workhorse of the theoretical literature (contributing to his Nobel prize).

**Other Comments Or Suggestions:**

1. sanity check: does the 'no negotiation' baseline match the Nordhaus-Yang results?
1. how does the current code compare to Nordhaus code?  Is it more modular, faster/slower?  (I had seen that there were RICE implementations in Julia a decade ago, but have lost track.)
1. it could be useful to plot policy functions as well (q.v. the comment in the Limitations section about interpretability).
1. Nordhaus appears in the references as both "W." and "W.D."  "Stern" and "AI" should be capitalised in the references.

**Other Strengths And Weaknesses:**

My biggest concern with the paper is the specification of binding negotiations.

In the 'real world', significant countries withdraw from their climate change commitments, partly as they do not trust others' commitment.  This, of course, is the standard stuff of game theory - and an essential part of the problem.

The original RICE paper (Nordhaus & Yang, 1996) noted that the international transfers required to support its first best solutions would swamp international transfers, making the solutions considered infeasible.

For research to make a significant contribution to this debate, it must at least address these issues.  The current paper, to my knowledge, does not.

I grade this paper as a 'weak accept': I think that this issue needs to be kept alive; I do not think that this significantly advances the debate (e.g. novel techniques, surprising results, useful policy insights), but it does help keep the issue alive.

**Questions For Authors:**

1. why are 27 fictitious countries used, rather than the full set?

**Relation To Broader Scientific Literature:**

Fine.

**Theoretical Claims:**

The paper makes no theoretical claims.

On this front, I was interested to know what - for example - the equilibrium concept is.  Overall, I felt that too many of the paper's details (e.g. the equilibrium concept, details of RL) were hidden in the appendix.

---

> ### Author Rebuttal · Authors · 2025-04-01
>
> We thank the reviewer for the incisive review of our submission. Below, we respond to the concerns raised.
>
> ### > does the 'no negotiation' baseline match the Nordhaus-Yang results?
>
> We can compare our results to Nordhaus’ RICE from 2010 [1]. The aforementioned has two results: baseline and optimal. The former consists of each region optimizing its own objective function and the latter of a cooperative mode with a shared objective function. We compare our results with RICE up until 2115, as that is the end year of our simulation. Looking at global temperature only for simplicity’s sake, the RICE2010 baseline and optimal temperature rises are 3.91 C and 2.92 C, respectively. Our no-negotiation baseline and basic club protocols result in increases of ~4.21 °C and ~2.97 °C, respectively. Exact matches are not expected as initial conditions differ: our rollout starts in 2015 (in line with DICE2016, which served as the basis for the implementation) and RICE2010 in 2005. Nonetheless, both no-negotiation and basic club yield temperature rises comparable to the RICE 2010 base and optimal trajectory, respectively.
>
> [1] ​​Nordhaus, W. D. (2010). Economic aspects of global warming in a post-Copenhagen environment. Proceedings of the NAS, 107(26), 11721-11726.
>
> ### >  Is [the code] more modular, faster/slower?
> While Python is likely slower than e.g. GAMS for model optimization, the runtime is bottlenecked by the deep RL agents, not the environment dynamics. Writing RICE-N in Python makes it simpler for the simulation code to interface with the RL agent code, which can be implemented using common Deep RL frameworks in Python and reduces barriers for ML researchers to contribute to the codebase, and extend it with novel modelling components. Finally, we are working on an efficient JAX implementation, which we plan to release alongside the paper and which will improve the runtime by multiple orders of magnitude.
>
> ### > My biggest concern with the paper is the specification of binding negotiations.
> In our current setup, agents cannot deviate from agreed-upon actions for the specified time lapse (5 years). While this assumption simplifies analysis and isolates the impact of the negotiation mechanism, it does not reflect the uncertainty and strategic mistrust present in real-world climate negotiations (e.g., countries withdrawing from agreements).
>
> Future work should explore the possibility of non-binding commitments, which would enable agents to engage in strategic communication or 'cheap talk,' thereby better capturing realistic negotiation dynamics [2]. We will add this discussion to the text.
>
> [2] Caparros, A. (2016). The Paris Agreement as a step backward to gain momentum: Lessons from and for theory. Revue d'économie politique, 126, 347.
>
>
> ### > Nordhaus & Yang issues related to international transfers >
> ​​The reviewer is correct to identify concerns related to international transfers at the scale required to implement RICE’s optimal pathways. For example, large-scale international transfers risk misappropriation without oversight [3]. However, in the time since Nordhaus and Yang’s 1996 comment on the practicality of international transfers, there has been a greater focus on equity and burden sharing [4], shifting the question from whether international transfers are possible to how do international transfers fit into a larger patchwork of climate finance instruments [5]. Adding an international transfers module is on the RICE-N development road map such that we can explore different means of financing mitigation.
>
> [3] Nest, M., et al. (2022). Corruption and climate finance. Brief Chr. Michelsen Institute U, 4, 14.
>
> [4] Landis, T. & Bernauer, T. (2012). Transfer payments in global climate policy. Nature Climate Change, 2(8), 628-633.
>
> [5] Pickering, J., et al. (2017). Special Issue: Managing Fragmentation and Complexity in the Emerging System of International Climate Finance. *International Environmental Agreements: Politics, Law and Economics*, 17(1), 1-16.
>
> ### > too many of the paper's details were hidden in the appendix.
> Thank you for this feedback. We will use the extra page of the camera-ready version to discuss more RL-related details in the main text.
>
> ### > it could be useful to plot policy functions as well
> Thank you for the suggestion. We will add to the final version.
>
> ### > Nordhaus appears in the references as both "W." and "W.D." "Stern" and "AI" should be capitalised in the references.
> Thank you for catching this. We have corrected this oversight.
>
> ### > why are 27 fictitious countries used, rather than the full set?
> Many individual countries lack comprehensive data making modeling unreliable. By aggregating countries into regions, the errors from lacking in data from these countries would be less impactful to the whole region. We believe, for a similar reason, RICE-2010 only includes 12 regions. Besides, 27-region setting makes bilateral negotiation computational feasible due to $O(n^2)$ complexity.

---

> > ### Comment · Reviewer_j6L5 · 2025-04-02
> >
> > Thank you.  Some thoughts in reply:
> >
> > 1. **Nordhaus-Yang baselines**: you mention that you are unable to compare the two models as N-Y starts in 2005, and you start in 2015.  Any paper I've seen that makes about its performance against benchmark/SOTA models necessarily compares the models on the same datasets.  Whether it's easier to run RICE-N on N-Y, or RICE on your data, I don't know.
> > 1. **comparison of code to N-Y**: I suspect that speed is not a big issue here, as we're analysing dynamics that take place over decades.  Instead, I am trying to understand how strong the argument is for a re-implementation of RICE is: the paper is weaker if it feels more like 'yet another...' rather than a novel contribution.
> > 1. **binding negotiations**: your comment about '5 years' confuses me: are agents myopically optimising?  This is why I originally asked about the equilibrium concept: for computations to be meaningful, we need to know _what_ is being computed: e.g. a Nash equilibrium, a subgame perfect Nash...
> > 1. **international transfers**: my initial view that this paper 'keeps the issue' alive continues - the paper seems to abstract from some of (what I regard as) the main reasons that this is a hard problem.
> > 1. **hidden in the appendix**: noted.
> > 1. **policy functions**: noted.
> > 1. **references**: noted.
> > 1. **27 countries**: again, I think that comparability to Nordhaus-Yang would be useful.  From this point of view, RICE-2010's 12 regions would be a natural baseline.

---

> > > ### Author Response · Authors · 2025-04-03
> > >
> > > Thank you for your continued engagement.
> > > ## 1. Comparison to RICE2010 baseline:
> > > To ensure the consistency of RICE-N’s no-negotiation baseline with Nordhaus2010, we will run the RICE-N no-negotiation baseline starting in 2005 as a sanity check. Since this will involve a significant recalibration effort, we can only promise this for the camera-ready deadline. We will also make sure to run the 12-region version.
> > >
> > > Taking a step back, we are not claiming our model is “SOTA”. In fact, there is no definition of SOTA among similar works, since we cannot evaluate the accuracy of prediction of temperature rise 100 years in the future. We compare other models with us for the purpose of sanity check.
> > >
> > > ## 2. Code comparison.
> > > Thank you for clarifying.
> > >
> > > ### 2.1
> > > The novelty of RICE-N is that it combines the RICE model with a multi-agent RL framework for climate-economic negotiations between regions.
> > > - Self-interested agents can negotiate, using “tools” such as trade and tariffs. This allows us to explore different negotiations that structure strategic interactions between regions differently.
> > > - Using deep RL algorithms to model agent policies avoids the need to manually specify agent policies for different negotiation protocols.
> > > ### 2.2
> > > To be clear: we do not claim novelty in our implementation of RICE in Python, for which we follow William Nordhaus’ 2010 implementation while we do add an extra international trading module. However, we didn’t emphasize our contribution here. A python interface is convenient, but not novel.
> > >
> > > ## 3. Binding negotiations:
> > > ### 5 year step:
> > > Sorry for the confusion. The agents are not myopic. Instead, RICE-N is a MARL agent-based model with per-agent independent model-free RL algorithms (A2C). Therefore, player strategies (agent policies) are not computed analytically, but independently learned iteratively by the agents with the goal of improving their individual returns, which correspond to their long-term aggregate rewards (see eqs 1 and 2).
> > > ### What is being computed:
> > > This agent-based model approach is relevant for complex settings [1], and deep RL is useful in settings where exact solutions may be intractable or difficult to qualify [2]. This is particularly helpful for RICE-N, which is meant to serve as a testbed for different negotiation protocols that can extend the environment with observations and actions beyond the original scope of RICE, such as punishment tariffs.
> > > ### The equilibrium concept:
> > > NY96 originally discusses pure strategy Nash equilibria. However, RICE-N departs from NY96 through the introduction of RL agents, international trade, tariffs and negotiation protocols. Importantly, since negotiation protocols can vary widely, no single equilibrium concept applies in all cases:
> > > - Equilibrium type: The negotiation protocol can give rise to the introduction of previously irrelevant equilibrium concepts, such as correlated equilibria from a stochastic protocol.
> > > - Augmenting observation/action spaces: The introduction of novel actions, such as punishment mechanisms (e.g. tariffs), can affect the equilibria, and in particular can create the possibility of self-enforcement through collective punishment for defection.
> > > - Commitment: Relaxing the commitment mask introduces cheap talk. A relevant solution concept here is the coalition-proof Nash equilibrium [3].
> > > - Information: The negotiation protocol can affect what information is public vs private.
> > >
> > > ## 4. Enforcement & International transfers:
> > > We fully agree that enforcement is a central component of any solution. We see this paper as a tool to help enable the study of different negotiation protocols and enforcement mechanisms. For example, future work could relax the formulation of the mask, structure self-enforcing negotiation protocols.
> > >
> > > Regarding transfers, we acknowledge that international transfers pose a hard problem with respect to the unequal distribution of funding requirements. RICE-N currently does not include international transfers, but rather opts for tariffs as a mechanism to incentivise mitigation. We are interested in including a climate finance module to explore potential avenues for climate finance, such as green bonds, technology sharing, and carbon pricing.
> > >
> > > [1] Bertsekas, Dimitri, and John N. Tsitsiklis. Neuro-dynamic programming. Athena Scientific, 1996.
> > >
> > > [2] Farmer, J. Doyne, et al. "A third wave in the economics of climate change." Environmental and Resource Economics 62 (2015): 329-357.
> > >
> > > [3] Bernheim, B. Douglas, Bezalel Peleg, and Michael D. Whinston. "Coalition-proof nash equilibria i. concepts." Journal of economic theory 42.1 (1987): 1-12.

---

### Official Review · Reviewer_aTjF · 2025-03-15

**Overall Recommendation:** 3

**Summary:**

The paper introduces and analyses a climate policy modelling framework for assessing the effect of different international agreemtns on future climate. It introduces RICE-N, a multi-region integrated assessment model that simulates global climate negotiations and agreements using multi-agent reinforcement learning to model strategic decision-making in climate policies. It develops and evaluates two negotiation protocols, Bilateral Negotiation and Basic Club which encourage cooperation among regions to mitigate climate change while balancing economic growth. The results indicate that these protocols have the potential to reduce global temperature rise give the particular simulation environment.

**Claims And Evidence:**

The claims in the submission are generally supported by the evidence, particularly through comparisons of negotiation protocols using multi-agent simulations and integrated assessment modelling. As the claims are made within the context of the RICE-N model, they are supported. Naturally, their applicability to real-world scenarios may be questioned but the paper does not claim to provide policy advice and outlines some possible unintended consequences in the Impact Statement.

**Essential References Not Discussed:**

I cannot comment on this.

**Experimental Designs Or Analyses:**

The experimental results provide some illustrative results for different policies though are not very extensive (e.g. in terms of comparing the effect of different variables, different assignment of regions, the computational considerations in the multi-agent RL inference). The authors mention that a more extensive sensitivity study would require a more efficient implementation of the methodology (e.g. in JAX).

**Methods And Evaluation Criteria:**

The proposed methods -- the use of MARL within the RICE-N integrated assessment model -- are suitable for studying strategic climate negotiations, as they allow agents to learn dynamic policies in a multi-region setting.

**Other Comments Or Suggestions:**

208, column 2: clarify that the link is to **Section** 6.

**Other Strengths And Weaknesses:**

The paper is well-written and includes an explanations of different aspects of climate negotiations, reinforcement learning methods, and integrated assessment modelling, making it accessible to a broad audience. Its originality lies in combining multi-agent reinforcement learning with climate-economic modelling, allowing for a more dynamic exploration of international climate agreements compared to static game-theoretic approaches. The availability of the implementation and the modularity of the it should make it easy for other users to try out their own approaches.

While the paper presents an interesting application of multi-agent reinforcement learning, its primary focus on climate economics and policy modelling may make it less suitable for a machine learning venue. To the best of my understanding, the technical ML contributions, such as improvements to MARL methods or novel learning dynamics, are limited, and the paper leans more toward applying existing ML techniques rather than advancing them. While it may be of interest to an ML audience due to potential future directions (improving scalability with JAX or more advanced RL methodology), I would suggest that the extensions are suitable for ML audience while this paper may be better received at a climate venue.

**Questions For Authors:**

Could you clarify what you mean by, the climate, economic and trade components are **loosely** coupled?

Most of the results contain some uncertainty estimates but I could not find the exact explanation of its origin. Could you explain or reference the specific part of the paper that details the source the stochasticity? Could you then give some intuition of why the uncertainty seems so small for most models and what changes to the model or the inference is likely to have a significant effect on the uncertainty?

**Relation To Broader Scientific Literature:**

The paper includes an extensive literature review on different aspects of the paper, including climate negotiations, climate negotiations and economic actions that affect climate outcomes, as well as some relevant RL literature.

**Theoretical Claims:**

There are no theoretical claims in the paper apart from the set up of the model that relates back to theoretical understanding of world economy, trade and international negotiations.

---

> ### Author Rebuttal · Authors · 2025-04-01
>
> Thank you for the thoughtful review and for highlighting the positive aspects, such as RICE-N’s suitability for studying strategic climate negotiations, the evidence-based claims, and the originality of our MARL application. We are pleased that you consider both the paper and the code accessible, and we hope that our response clarifies any outstanding concerns.
>
> ### > Suitability for ICML.
>
> Thank you for voicing your concerns. We believe that the application of MARL to a climate economic negotiation integrated assessment model provides a relevant methodological contribution to the Application-Driven Machine Learning track of ICML. Beyond its contribution to topics mentioned in the call for papers such as social sciences, and sustainability and climate for which the ML community demonstrates interest through concurrent work on climate investment [1], as well as the many potential extensions that are interesting to the ML community, such as the integration of LLMs into negotiation [2], this work also provides a useful testbed for topics of interest to MARL researchers, such as studying multi-party cooperation between AI agents in sequential social dilemmas [3].
>
> [1] Hou, X., et al. (2025). InvestESG: A multi-agent reinforcement learning benchmark for studying climate investment as a social dilemma. In ICLR 2025.
>
> [2] Vaccaro, M., et al. (2025). Advancing AI negotiations: New theory and evidence from a large-scale autonomous negotiations competition. arXiv preprint arXiv:2503.06416.
>
> [3] Leibo, J. Z., et al. (2017). Multi-agent reinforcement learning in sequential social dilemmas. In Proceedings of the 16th AAMAS. ACM.
>
> ### > Additional experimental analysis on effects of variables.
>
> Since the space of possible configurations is large, we perform a sensitivity analysis over a subset of economically relevant parameters, namely the discount factor, welfare loss weight, consumption substitution rate and relative preference for domestic goods. This analysis showcases the robustness of our findings. We will add it to the appendix and reference it in the main text. We plot heatmaps of the analysis here: https://imgur.com/a/W0194Ej . We show the percentage change in outcome variables of interest across different scenarios when critical model parameters are perturbed by a multiplication factor ranging from 0.96 to 1.04. The max percentage change is 3.16% while mean is -0.22% and medium is -0.36%. We thus conclude the dynamics are stable, corresponding to changes in critical model parameters.
>
> We will also include a discussion on the computational complexity of RICE-N (please see our response to QTAV).
>
> ### > 208, column 2: clarify that the link is to Section 6
>
> Thank you for spotting this.
>
> ### > possible unintended consequences
>
> We wish to highlight the importance of qualifying these claims by discussing the potential unintended consequences of climate clubs being implemented without redistributive financing and technology sharing. Uniform tariffs on developing countries with lower mitigation rates would effectively serve as a tax on less developing countries, which we do not advocate for [4,5]. Our goal is to create a simulation framework where these dynamics and alternative policies can be explored and cross compared. We will give an account of these points in the main text of the camera-ready version.
>
> [4] Goldthau, A., & Tagliapietra, S. (2022). How an open climate club can generate carbon dividends for the poor. Bruegel-Blogs.
>
> [5] Perdana, S., & Vielle, M. (2022). Making the EU Carbon Border Adjustment Mechanism acceptable and climate friendly for least developed countries. Energy Policy, 170, 113245.
>
> ### > loosely coupled
>
> To clarify: this comment refers to the structure of the code, not any functional difference. We mean that we try to reduce as much as possible the number of assumptions that each component must make about the other components, and make the necessary dependencies between dynamics both explicit and local [6]. This makes extensions to the existing codebase much easier to implement, which makes testing new ideas, such as novel negotiation protocols, much easier.
>
> [6] Leymann, F. (2016, September 5–7). Loose coupling and architectural implications. Keynote address presented at the ESOCC, Vienna, Austria.
>
> ### > uncertainty estimates
>
> The uncertainties arise from the stochasticity of the learned policy as we sample actions from the policy's distribution during evaluation. We estimate uncertainty across 50 rollouts with varying seeds (see Section 6, paragraph “Experimental Setup”). The relatively small uncertainty shows that the agents have learned robust policies. An increase could indicate that the learned policy is less reliable, potentially due to insufficient training or the model's inability to generalize effectively. Adding uncertainty to the environment (e.g., the climate component) or relevant parameters (see sensitivity analysis) would likely result in an increase of the uncertainty.

---

> > ### Comment · Reviewer_aTjF · 2025-04-05
> >
> > Thank you for your responses. I particularly appreciate the clarification regarding the suitability for the venue and I'm happy to update my recommendation.
> >
> > I would suggest adding a more careful discussion of uncertainty estimates and sensitivity analysis as pointed out by other reviewers to the final version of the paper.

---

> > > ### Author Response · Authors · 2025-04-06
> > >
> > > Thank you for your kind feedback and for updating your recommendation. We appreciate your constructive suggestion and will incorporate a more detailed discussion on uncertainty estimates and sensitivity analysis in the final version of the paper, as recommended.

---

### Official Review · Reviewer_qtav · 2025-03-20

**Overall Recommendation:** 4

**Summary:**

The paper introduces RICE-N, a multi-region integrated assessment model designed to simulate global climate negotiations, agreements, and long-term cooperation using multi-agent reinforcement learning (MARL). The model extends the Regional Integrated Model of Climate and Economy (RICE) by incorporating negotiation protocols and international trade dynamics. The authors propose two negotiation protocols: Bilateral Negotiation and Basic Club, inspired by real-world climate policy mechanisms like Climate Clubs and the Carbon Border Adjustment Mechanism (CBAM). The main findings are that both negotiation protocols significantly reduce temperature growth and carbon emissions compared to a no-negotiation baseline, with only a minor drop in production. The Basic Club protocol, in particular, achieves a balance between emissions reduction and economic growth, outperforming the no-negotiation baseline in the long term. The paper also highlights the importance of equitable burden-sharing in climate agreements, as measured by the Gini Index.

**Claims And Evidence:**

The claims made in the paper are generally supported by clear and convincing evidence. The authors provide detailed simulations and comparisons between different negotiation protocols and baselines, showing the impact on global temperature, carbon emissions, and economic output. The use of MARL to model strategic behavior in climate negotiations is well-justified, and the results are presented with appropriate statistical measures (e.g., mean ± 1.96 standard error). However, the paper could benefit from more detailed sensitivity analyses to demonstrate the robustness of the results to different parameter settings or model assumptions.

**Essential References Not Discussed:**

The paper adequately covers the relevant literature, but it could benefit from a more detailed discussion of recent work on MARL in climate-related applications. For example, recent papers on MARL for energy systems optimization or climate policy design could provide additional context for the use of MARL in this domain. Additionally, the paper could discuss more recent developments in climate clubs and carbon border adjustment mechanisms, which have been the subject of ongoing policy debates.

**Experimental Designs Or Analyses:**

The experimental design is sound, with clear comparisons between different negotiation protocols and baselines. The authors train five models (Basic Club, Bilateral Negotiation, no negotiation, maximum mitigation, and minimum mitigation) and evaluate them over 50 rollouts to ensure statistical robustness. The results are presented with appropriate confidence intervals, and the authors discuss the implications of their findings in detail. One potential limitation is the lack of sensitivity analysis to different parameter settings, which could strengthen the robustness of the results.

**Methods And Evaluation Criteria:**

The proposed methods and evaluation criteria are appropriate for the problem at hand. The use of MARL to model strategic interactions in climate negotiations is innovative and well-suited to the complex, dynamic nature of global climate cooperation. The evaluation criteria, including global temperature anomaly, carbon emissions, and economic output, are standard metrics in climate-economic modeling. The inclusion of the Gini Index to measure inequality in emission reduction costs and consumption adds a valuable dimension to the analysis, addressing the equity concerns often raised in climate negotiations.

**Other Comments Or Suggestions:**

1. The paper is well-written and clearly organized, but it could benefit from a more detailed discussion of the limitations of the model, particularly in terms of its assumptions about agent behavior and the real-world applicability of the negotiation protocols.

2. The authors should consider adding a discussion of the potential policy implications of their findings, particularly in light of ongoing international climate negotiations.

**Other Strengths And Weaknesses:**

Strengths:

1. The paper addresses a critical and timely issue in climate policy, namely the challenge of achieving global cooperation on climate change mitigation.

2. The use of MARL to model strategic behavior in climate negotiations is innovative and well-executed.

3. The inclusion of international trade dynamics and negotiation protocols adds realism to the model and provides valuable insights into the design of effective climate agreements.

4. The paper provides a comprehensive analysis of the equity implications of different negotiation protocols, which is often overlooked in climate-economic modeling.

Weaknesses:

1. The paper could benefit from a more detailed sensitivity analysis to demonstrate the robustness of the results to different parameter settings or model assumptions.

2. The discussion of the real-world applicability of the Basic Club protocol could be expanded, particularly in light of ongoing policy debates on carbon border adjustment mechanisms.

3. The paper could provide more details on the computational requirements of the MARL approach, particularly for large-scale simulations with many regions.

**Questions For Authors:**

## Sensitivity Analysis
**Question:** Have the authors conducted a sensitivity analysis to test the robustness of the results to different parameter settings or model assumptions? If so, could they provide more details on the findings?

**How this affects my evaluation:** If the authors can demonstrate that the results are robust to different parameter settings, it would strengthen the validity of their conclusions.

## Real-World Applicability
**Question:** How do the authors see the Basic Club protocol being implemented in real-world climate negotiations, particularly in light of ongoing debates on carbon border adjustment mechanisms?

**How this affects my evaluation:** A more detailed discussion of the real-world applicability of the Basic Club protocol would enhance the practical relevance of the paper.

## Computational Requirements
**Question:** What are the computational requirements of the MARL approach, particularly for large-scale simulations with many regions?

**How this affects my evaluation:** Understanding the computational requirements would help assess the scalability of the approach and its potential for real-world applications.

**Relation To Broader Scientific Literature:**

The paper is well-situated within the broader scientific literature on climate-economic modeling and game-theoretic approaches to climate negotiations. The authors draw on prior work in integrated assessment models (IAMs) like DICE and RICE, as well as recent advances in MARL. The paper extends this literature by incorporating negotiation protocols and international trade dynamics, which are critical for modeling real-world climate agreements. The use of MARL to model strategic behavior in climate negotiations is a novel contribution that bridges the gap between climate economics and machine learning.

**Theoretical Claims:**

The paper does not present any formal theoretical claims or proofs, so there are no theoretical issues to evaluate. The focus is on empirical results from simulations, which are well-documented and supported by the data.

---

> ### Author Rebuttal · Authors · 2025-04-01
>
> We thank the reviewer for their favorable assessment regarding the innovativeness and suitability of our chosen approach, as well as the constructive feedback. Below, we carefully address any outstanding concerns:
>
> ### > detailed sensitivity analysis
>
> Thank you for this suggestion. We select parameters based on economic theory, which are the discount factor, welfare loss weight, consumption substitution rate and relative preference for domestic goods. This analysis showcases the robustness of our findings. We will add it to the appendix and reference it in the main text. We plot heatmaps of the analysis here: https://imgur.com/a/W0194Ej . In the heatmap, we show the percentage change in variables of interest (including Temperature, Carbon Emissions, GDP) across different scenarios when critical model parameters are perturbed by a multiplication factor ranging from 0.96 to 1.04. The max percentage change is 3.16% while mean is -0.22% and medium is -0.36%. We thus conclude the dynamics are stable, corresponding to changes in critical model parameters.
>
> ### > real-world applicability of the Basic Club protocol
>
> Basic Club and carbon border adjustment mechanisms (CBAM) fight carbon leakage by setting the strength of the tariff proportionally to the difference in the cost of carbon between the exporting and importing countries. Basic Club relies on a uniform tariff [1]. In contrast, CBAM targets specific goods, which can exacerbate carbon leakage by leaving large swaths of emissions unaccounted for, such as those produced for non-EU exports. Climate clubs can go further than what is modeled in RICE-N at the moment, hence the term “Basic”. Uniform tariffs, coupled with technology sharing and redistribution, are theorized to be more effective at curtailing carbon leakage [2]. Future work should implement CBAM in RICE-N, which requires disaggregating production and trade by sector to allow for targeted tariffs. This is currently a work in progress and high on our agenda.
>
> [1] Overland, I., & Huda, M. S. (2022). Climate clubs and carbon border adjustments: A review. *Environmental Research Letters*, 17(9), 093005.
>
> [2] Tarr, D. G., Kuznetsov, D. E., Overland, I., & Vakulchuk, R. (2023). Why carbon border adjustment mechanisms will not save the planet but a climate club and subsidies for transformative green technologies may. *Energy Economics*, 122, 1066
>
> ### > What are the computational requirements of the MARL approach, particularly for large-scale simulations with many regions?
>
> The computational complexity of our MARL approach is driven by the number of regions N (i.e., agents). Since each agent’s action space scales linearly with N, the total action space across all agents grows quadratically (O(N²)). However, the number of agents in our setting is naturally bounded by the number of countries on the planet. Currently, training 27 agents for 100 thousand episodes takes approximately 3 hours on a 30 CPU cluster. To improve runtime, we are exploring more efficient implementations using JAX-based acceleration and model parallelism, which would enable large-scale sensitivity analyses and experiments.
>
> ### >  more detailed discussion of recent work on MARL in climate-related applications.
>
> We extend the section on MARL in the appendix with some more references on MARL applied to real-world problems such as:
>
> Hou, X., et al. (2025). InvestESG: A multi-agent reinforcement learning benchmark for studying climate investment as a social dilemma. In *Proceedings of the Thirteenth International Conference on Learning Representations (ICLR)*.
>
> May, R., & Huang, P. (2023). A multi-agent reinforcement learning approach for investigating and optimising peer-to-peer prosumer energy markets. *Applied Energy*, 334, 120705.
>
> Wang, X., et al. (2020). Large-scale traffic signal control using a novel multiagent reinforcement learning. *IEEE Transactions on Cybernetics*, 51(1), 174–187.
>
> ### > discussion of the potential policy implications
>
> The policy impacts of climate clubs (and border adjustment mechanisms for that matter) depends on their implementation. Without some degree of redistribution and technology sharing, they risk acting as a tax on carbon locked developing countries [3; 4]. A Loss and damage fund can mitigate that risk [5].
>
> [3] Goldthau, A., & Tagliapietra, S. (2022). How an open climate club can generate carbon dividends for the poor. Bruegel-Blogs.
>
> [4] Perdana, S., & Vielle, M. (2022). Making the EU Carbon Border Adjustment Mechanism acceptable and climate friendly for least developed countries. *Energy Policy*, 170, 113245.
>
> [5] Boyd, E., et al. (2021). Loss and damage from climate change: A new climate justice agenda. *One Earth*, 4(10), 1365–1370.
>
> ### > more detailed discussion of the limitations of the model.
>
> Thank you for this suggestion. We will add a clarifying paragraph to the main text describing key assumptions around binding commitments, region preferences, and power imbalances.

---

### Decision · Program_Chairs · 2025-05-01

**Decision:**

Accept (poster)

**Comment:**

The paper proposed a multi-agent reinforcement learning (MARL) approach to extend the Nordhaus Regional Integrated Model of Climate and Economy (RICE) model of global climate negotiations, agreements, and long-term cooperation on cutting greenhouse gases emissions. Two novel negotiation protocols were proposed: Bilateral Negotiation and Basic Club. The proposed approach and protocols take into account many realistic components such as climate dynamics and economic dynamics.

Reviewers ntBn, qtav and aTjF praised the novelty of incorporating MARL into negotiation protocols, and evaluating outcomes using Gini indexes. Reviewer aTjF found the paper well written. Reviewers j6L5 and aTjF  found the claims to be evidenced by the paper.

Reviewer ntBn noted that results were shown mostly at global macro level and not at regional level. Reviewer j6L5 discussed the limitation of not specifying negotiations to be binding. Reviewer aTjF found a climate focused venue could be more suitable for the paper (claim defended by the authors).

Based on the reviews and average score, I recommend accepting the paper.